# Modeling the Formation and Composition of Secondary Organic Aerosol from Diesel Exhaust Using Parameterized and Semi-Explicit Chemistry and Thermodynamic Models

Sailaja Eluri[1], Christopher D. Cappa[2], Beth Friedman[3], Delphine K. Farmer[3], and Shantanu H. Jathar[1]

[1] Department of Mechanical Engineering, Colorado State University, Fort Collins, CO, USA, 80523

[2] Department of Civil and Environmental Engineering, University of California Davis, Davis, CA, USA, 95616

[3] Department of Chemistry, Colorado State University, Fort Collins, CO, USA, 80523

Correspondence to: Shantanu H. Jathar (shantanu.jathar@colostate.edu)

**Abstract**

Laboratory-based studies have shown that combustion sources emit volatile organic compounds that can be photo-oxidized in the atmosphere to form secondary organic aerosol (SOA). In some cases, this SOA can exceed direct emissions of primary organic aerosol (POA). Jathar et al. (2017) recently reported on experiments that used an oxidation flow reactor (OFR) to measure the photochemical production of SOA from a diesel engine operated at two different engine loads (idle, load), two fuel types (diesel, biodiesel) and two aftertreatment configurations (with and without an oxidation catalyst and particle filter). In this work, we used two different SOA models, the volatility basis set (VBS) model and the statistical oxidation model (SOM), to simulate the formation and composition of SOA for those experiments. Leveraging recent laboratory-based parameterizations, both frameworks accounted for a semi-volatile and reactive POA; SOA production from semi-volatile, intermediate-volatility, and volatile organic compounds (SVOC, IVOC and VOC); $NO_X$-dependent parameterizations, multigenerational gas-phase chemistry; and kinetic gas/particle partitioning. Both frameworks demonstrated that for model predictions of SOA mass to agree with measurements across all engine load-fuel-aftertreatment combinations, it was necessary to model the kinetically-limited gas-particle partitioning in OFRs as well as account for SOA formation from IVOCs, which were on average found to account for 70% of the model-predicted SOA. Accounting for IVOCs however resulted in an average under-prediction of 28% for OA atomic O:C ratios. Model predictions of the gas-phase organic compounds (resolved in carbon and oxygen space) from the SOM compared favorably to gas-phase measurements from a Chemical Ionization Mass Spectrometer (CIMS), substantiating the semi-explicit chemistry captured by the SOM. Model-measurement comparisons were improved on using vapor wall-loss corrected SOA parameterizations. As OFRs are increasingly used to study SOA formation and evolution in laboratory and field environments, models such as those developed in this work can be used to interpret the OFR data.

## 1 Introduction

Combustion-related aerosols are an important contributor to urban and global air pollution and have impacts on climate (Pachauri et al., 2014) and human health (Anderson et al., 2012). While direct particle emissions from combustion sources are dominated by primary organic aerosol (POA) and black carbon (Bond et al., 2004), these sources also emit more volatile organic compounds (VOCs) that can photochemically react in the atmosphere to form secondary organic aerosol (SOA) (Robinson et al., 2007). SOA production from combustion emissions is poorly understood and not very well represented in models in terms of its precursors, gas-particle partitioning, composition, and properties (Fuzzi et al., 2015). Atmospheric models frequently under-predict SOA mass concentrations during strong photochemical episodes in urban areas (Jathar et al., 2017b), which likely highlights the challenge in modeling the SOA contributions from urban, combustion-related emissions (Ensberg et al., 2014).

Diesel-powered sources, which are an important source of air pollution at urban and regional scales, emit precursors that form SOA in the atmosphere (Gentner et al., 2016). Robinson et al. (2007) found that photochemical processing of exhaust emissions from a small off-road diesel engine led to SOA production and doubled the primary aerosol mass over a few hours in an environmental chamber. Chirico et al. (2010) and Gordon et al. (2014) performed similar chamber experiments on tailpipe emissions from in-fleet, on-road diesel vehicles run on chassis dynamometers. Both found SOA production that was roughly consistent with the findings from Robinson et al. (2007). They additionally found that the use of aftertreatment devices (diesel oxidation catalysts and diesel particulate filters) substantially reduced SOA production (mimicking the reduction in primary aerosol emissions) but observed some SOA production during cold starts and/or regeneration events when the proper functioning of the aftertreatment devices was limited. Furthermore, Gordon et al. (2014) found negligible differences in the SOA formation between diesel and biodiesel fuel. To access longer equivalent photochemical aging timescales compared to typical chamber experiments, Tkacik et al. (2014) measured SOA formation using an oxidation flow reactor (OFR) from air sampled from a highway tunnel in Pittsburgh, PA used by both on-road gasoline and diesel vehicles. OFRs use high concentrations of atmospheric oxidants, e.g. hydroxyl radicals, to achieve long exposures on short actual timescales; further discussion is provided below. Tkacik et al. (2014) measured much stronger SOA formation compared to chambers (SOA: POA was 10:1) over photochemical exposures equivalent to 2 to 3 days, but found that the SOA was lost, or destroyed, as the mixture continued to age over the timescale of a week. Recently, Jathar et al. (2017a) performed experiments using an oxidation flow reactor to measure the photochemical production of SOA from an off-road diesel engine operated at various engine load, fuel, and aftertreatment configurations. Jathar et al.

(2017a) found that efficient combustion at higher engine loads and removal of SOA precursors by aftertreatment systems reduced SOA production by factors of 2 to 10. The only exception was that the aftertreatment system did not seem to reduce SOA production at idle loads possibly because the exhaust temperatures were low enough to limit removal of SOA precursors in the oxidation catalyst. Overall, these studies indicate that diesel exhaust contributes to atmospheric SOA production, although the precise production of SOA varies across dimensions of photochemical age, engine duty cycle, use of alternative fuels, and aftertreatment devices.

Oxidation flow reactors are being used to study the photochemical production of SOA from both anthropogenic (e.g., Ortega et al. (2016)) and natural (e.g., Palm et al. (2016)) sources. Most OFRs used for SOA studies are 10 to 15 L, flow-through metal reactors with lamps that can produce high concentrations of atmospheric oxidants to simulate photochemical processing (e.g., Lambe et al. (2011)). Flows through an OFR allow for residence times between one and four minutes, but given the high oxidant concentrations OFRs can simulate several weeks of photochemistry. OFRs have three distinct advantages over environmental chambers. First, OFRs are smaller in size and easier to operate than environmental chambers, which allows for shorter experiments and makes them ideal for field deployments (Palm et al., 2016;Simonen et al., 2017). Second, production of high oxidant concentrations in OFRs allows for much longer photochemical exposures (~factor of 10) than those possible with chambers (Lambe et al., 2011). Third, due to their flow-through nature, OFRs have shorter residence times than conventional chambers (~1-4 minutes) and hence are less susceptible to gas and particle losses that can influence SOA formation (Zhang et al., 2014;Krechmer et al., 2016). Despite those advantages, there are concerns that the accelerated chemistry and limitations to gas/particle partitioning may affect the formation and composition of SOA in OFRs, which calls into question their relevance in understanding SOA formation in the real atmosphere (Palm et al., 2016;Jathar et al., 2017a;Ahlberg et al., 2017). For example, short residence times and/or small condensation sinks from preexisting aerosol may not allow for complete condensation of SOA vapors (Lambe et al., 2015). Similarly, high oxidant concentrations in OFRs may lead to molecules undergoing a greater number of reactions in the gas-phase before condensing, including reactions that lead to fragmentation and formation of higher volatility products (Kroll et al., 2009). Both effects will typically suppress SOA production. With the increased use of OFRs, there is a need to develop and use modeling tools that can account for fragmentation reactions and kinetic gas/particle partitioning. This will allow for a more accurate interpretation of OFR data and facilitate translation of OFR results to the real atmosphere.

Models used to simulate the photochemical production of SOA from VOCs in combustion emissions have
traditionally used the two-product (Odum et al., 1996) or the more generalized n-product volatility basis
set (VBS) framework (Donahue et al., 2006a). In this framework, VOC oxidation products are lumped
into volatility bins based on their effective saturation concentrations (C*) and where the saturation
concentration determines the partitioning of the products between the gas and particle phases (Pankow,
1994). This framework has been widely used in both box (Dzepina et al., 2009;Hodzic et al., 2010;Jathar
et al., 2014a;Hayes et al., 2015) and three-dimensional (Murphy and Pandis, 2009;Tsimpidi et al.,
2009;Jathar et al., 2011;Ahmadov et al., 2012;Konovalov et al., 2015) models to simulate the chemistry
and gas/particle partitioning of SOA. While this framework offers a simple and computationally efficient
scheme to model SOA formation, the use of volatility alone neither tracks the molecular composition, nor
informs the continued multi-generational chemistry that will determine the atmospheric evolution and
properties of SOA. As a result, volatility-based models have been challenged in leveraging observations
of the elemental composition of SOA (e.g., atomic O:C ratios) that have become possible through the use
of the aerosol mass spectrometer (AMS) to constrain parameterizations or test model predictions. Further,
most volatility-based models have employed *ad hoc* parameterizations to model multi-generational
chemistry that do not account for fragmentation reactions (Robinson et al., 2007) and possibly double
count SOA formation (Jathar et al., 2016). Therefore, there is a demand to develop models that can
provide an improved representation of the chemistry that governs the formation, composition, and
properties of SOA.

Previously, volatility-based SOA models have been used to predict photochemical production of SOA
from motor vehicle exhaust (Robinson et al., 2007;Jathar et al., 2014b;Tkacik et al., 2014). These
modeling studies have shown that speciated SOA precursors such as long alkanes ($C_{6-12}$) and single-ring
aromatics (e.g., benzene, toluene) explain less than 20% of the observed SOA and have argued that the
remainder of the SOA (~80%) arises from the photooxidation of typically unspeciated organic
compounds. These unspeciated compounds, also referred to as intermediate volatility organic compounds
(IVOCs), are likely species with carbon numbers larger than 12 and appear as an unresolved complex
mixture on using traditional gas chromatography mass spectrometry (GC-MS) techniques (Presto et al.,
2011). Early estimates of IVOC emissions and their SOA potential have significantly improved
predictions of the SOA formed from diesel exhaust (Jathar et al., 2014b) and have broadly improved OA
model performance in three-dimensional large-scale models (Murphy and Pandis, 2009;Pye and Seinfeld,
2010;Jathar et al., 2011;Tsimpidi et al., 2009). Consider as an example that Zhao et al. (2015), using a
thermal desorption GC-MS to provide detailed speciation of the carbon-number resolved linear, branched,
and cyclic alkane IVOCs in diesel exhaust, found that these species accounted for up to 60% of the non-
methane organic gas emissions. While IVOCs have been recognized as an important class of SOA
precursors for diesel (and even for gasoline and biomass burning) sources, updated emissions and
speciation estimates from Zhao et al. (2015) have not yet been used to explain observations of
photochemically produced SOA from diesel exhaust.

Recently, several model frameworks have been developed to improve the representation of SOA
formation, considering dimensions of SOA beyond just volatility. The statistical oxidation model (SOM)
developed by Cappa and Wilson (2012) is one such example, although volatility remains an important
consideration. The SOM is a semi-explicit, parameterizable mechanism that uses a two-dimensional
carbon-oxygen grid to simulate the multigenerational chemistry and gas/particle partitioning of organic
compounds. Although the SOM does not explicitly track or specify the product species composition (e.g.,
functional groups), the carbon- and oxygen-number representation provides adequate detail to represent
many key atmospheric processes, e.g., reactions with oxidants, formation of functionalized products,
scission of carbon backbones or fragmentation, surface and condensed-phase chemistry and gas/particle
partitioning. The SOM has been used to interpret chamber experiments (Zhang et al., 2014;Cappa et al.,
2013;Cappa and Wilson, 2012) and was recently integrated into a chemical transport model (Jathar et al.,
2015) to examine the influence of multigenerational aging (Jathar et al., 2016) and chamber-based vapor
wall losses (Cappa et al., 2016) on ambient concentrations and properties of OA. The two-dimensional
VBS (2D-VBS) of Donahue et al. (2011) and the carbon-polarity grid of Pankow and Barsanti (2009) are
examples of similar frameworks. These more sophisticated models (i.e., SOM, 2D-VBS, carbon-polarity
grid) have not yet been employed to study SOA formation from complex mixtures such as combustion
emissions.

To summarize, combustion sources such as diesel-powered sources emit precursors that can photooxidize
in the atmosphere to produce SOA. This SOA production is dependent not only on the precursor
composition (that could vary by combustion mode and fuel type) and photochemical age, but also
experimental artifacts (e.g., short condensation timescales) introduced by oxidation flow reactors. Hence,
there is a need to develop and apply sophisticated, yet computationally efficient, numerical models to
simulate and study SOA formation from combustion emissions. In this work, we applied two SOA model
frameworks that vary in sophistication (VBS and SOM) to simulate the photochemical production of SOA
in an OFR from diesel exhaust. The models were evaluated by comparing model predictions (OA and
O:C) to the recent measurements made by Jathar et al. (2017a) where SOA production was quantified for
different photochemical ages under varying engine loads, fuels, and aftertreatment devices. The model-
measurement comparison, along with sensitivity simulations, highlights the importance of modeling the
kinetic gas/particle partitioning of SOA in OFRs, the contribution of IVOCs to the total SOA production,
and the ability of the SOM to accurately track the composition of SOA.

**2 Methods**
**2.1 Experiments and Data**
Jathar et al. (2017a) performed photooxidation experiments using an OFR to measure SOA production
from the exhaust of a 4.5L, John Deere diesel engine. The stock engine met Tier 3 emissions standards for
off-road diesel engines. The OFR used therein was described in detail by Friedman et al. (2016) and the
experimental setup and OA measurements from these experiments were described in detail by Jathar et al.
(2017a). We briefly summarize the experimental setup, measurements, and findings from Jathar et al.
(2017a). The engine was run at two different loads (idle and 50% load) with two different fuels (diesel
and biodiesel), and with and without an aftertreatment system. The aftertreatment system included a diesel
oxidation catalyst (DOC) to oxidize CO and THC and a diesel particle filter (DPF) to trap fine particles.
Diesel exhaust was diluted by a factor of 45-110 before entering the OFR. The intensity of the mercury
lamps (at wavelengths of 185 and 254 nm) inside the OFR was varied to produce different hydroxyl
radical (OH) concentrations and simulate different photochemical exposures. The OFR had a residence
time of 100 s. A suite of instrumentation was used to measure gas- ($CO_2$, CO, total hydrocarbons, $NO_x$,
$O_2$, oxygenated organic compounds) and particle- (aerosol size and composition) phase concentrations. A
total of fourteen experiments (see Table 1 for more details) were performed at varying engine loads and
with varying fuels and aftertreatment configurations. The OH exposure was varied between 0 and a
maximum of $9.2 \times 10^7$ molecules-hr cm$^{-3}$ (equivalent to 2 days of photochemical aging at an OH
concentration of $1.5 \times 10^6$ molecules cm$^{-3}$). On average, each experiment included measurements at six to
seven different photochemical exposures. The mass concentrations and elemental composition of the POA
(measured when OFR lights were off) and SOA (at varying OH exposures) were measured by a high-
resolution aerosol mass spectrometer (HR-AMS). In addition to the measurements reported by Jathar et al.
(2017a), the gas-phase concentrations of oxygenated organic compounds were measured by an acetate
reagent ion-based chemical ionization mass spectrometer (CIMS) (Link et al., 2016). At all engine
configurations, SOA production exceeded the POA emissions after the equivalent of a few hours of
atmospheric photochemical aging. SOA production was particularly strong at idle (or less fuel-efficient)
engine loads and/or when exhaust temperatures were low and proper functioning of the aftertreatment
devices was limited. Further, POA emissions and SOA production were nearly identical between diesel
and biodiesel fuels. A synopsis of experiments performed and the total hydrocarbons (THC), which
includes all SOA precursors, POA, SOA, O:C, OH, and size distribution data are presented in Table 1.

Although the diesel exhaust was diluted with clean air to produce atmospherically-relevant concentrations
of POA, the initial THC, CO, and $NO_X$ concentrations in the OFR were still quite high. Peng and Jimenez
(2017), using a detailed gas-phase model, argued that the high external OH reactivity from high THC,
CO, and $NO_X$ concentrations might lead to non-OH chemistry in the OFR and NO could quickly be
consumed in the OFR leading to low NO conditions for SOA formation. Peng and Jimenez (2017)
quantified the potential influence of NO on the oxidation chemistry by calculating the ratio of the reactive
flux of the peroxy radicals with NO to the reactive flux of the peroxy radicals with $HO_2$ ($r_{RO2+NO}/r_{RO2+HO2)}$.
A ratio greater than 1 was considered as "high NO" while a ratio less than 1 was considered "low NO".
For the relative humidity, photon flux, initial NO, and external OH reactivity values in Jathar et al.
(2017a), the model of Peng and Jimenez (2017) predicted that the OFR mostly ran in a high NO mode at
all photochemical exposures when the engine was run at load conditions or with an aftertreatment device
in place. However, the model predicted that the OFR mostly ran in a low NO mode especially at the high
photochemical exposures when the engine was run at idle conditions and without an aftertreatment device
(i.e., Idle-Diesel-None and Idle-Biodiesel-None). The $r_{RO2+NO}/r_{RO2+HO2}$ ratio and low versus high NO
mode for each photon flux-experiment combination is listed in Table S1. Based on these results, we
accordingly used the low and high $NO_X$ parameterizations to perform the model simulations.

The model of Peng and Jimenez (2017) also indicated that most of the experiments performed by Jathar et
al. (2017) were run under, in what Peng and Jimenez (2017) refer to as, 'risky' or 'bad' conditions. These
conditions refer to situations in the OFR where the initial NO concentrations and external OH reactivity
are high enough to suppress OH exposure and lead to non-tropospheric photolysis at 185 and 254 nm,
which could compete with OH exposure to determine the fate of the SOA precursors and its oxidation
products. Such conditions could be avoided by ensuring low initial NO concentrations and external OH
reactivity that for combustion emissions would require substantial dilution with clean air before they are
oxidized in the OFR. Future studies on combustion sources should be cognizant of this fact to avoid
artifacts linked to non-tropospheric photolysis of organic compounds in OFRs.

**2.2 Organic Aerosol Models**
In this work, we used two different OA models to predict the mass concentrations and chemical
composition of SOA and compare predictions against the SOA measurements from Jathar et al. (2017a)
and Friedman et al. (2017). In this section, we briefly describe the two model frameworks, namely the
Volatility Basis Set (VBS) and the Statistical Oxidation Model (SOM), used to simulate the coupled
chemistry, thermodynamic properties, and kinetic gas/particle partitioning of OA. Neither model
accounted for photolysis of organic compounds in the gas phase at 185 or 254 nm, which may need to be
considered in the future when modeling the OFR chemistry from combustion emissions. The VBS model
was chosen as it is widely used in contemporary air quality models; the SOM was chosen to examine the
influence of improved representation of OA processes (e.g., fragmentation reactions) on model
predictions.

**2.2.1 Volatility Basis Set**
The Volatility Basis Set model, developed by Donahue et al. (2006b), is a parameterizable model that
allows for a volatility-based representation of the coupled chemistry, thermodynamic properties, and
gas/particle partitioning of OA. The VBS uses logarithmically spaced so-called basis sets based on the
effective saturation concentration ($C^*$); $C^*$ of a species determines the partitioning between the gas and
particle phases (Pankow, 1994). In the VBS model, organic precursors were allowed to react with OH to
yield a unique product distribution in $C^*$ space that represented stable first-generation products.
Subsequent multi-generational gas-phase oxidation, or so-called 'aging,' of the VBS products was
modeled using the scheme of Robinson et al. (2007). In this scheme the product species are allowed to
react with OH and yield a product with a $C^*$ that is an order of magnitude lower than the direct precursor,
to a lower limit $C^*$ of $10^{-2}$ µg m$^{-3}$. This scheme did not consider fragmentation reactions. The following
equations were used to represent the organic precursor oxidation (equation 1) and subsequent reaction and
formation of products from the precursor oxidation and aging reactions (equation 2):
$$\frac{dV}{dt} = -k_{OH}[V][OH] \tag{1}$$
$$\frac{dc_j^{g+p}}{dt} = \alpha_j k_{OH}[V][OH] + \beta\, k_{OH,aging}[C_{j+1}^g][OH] - \gamma\, k_{OH,aging}[C_j^g][OH] \tag{2}$$
where $V$ is the gas-phase concentration of a generic organic precursor (µg m$^{-3}$; includes VOCs, IVOCs
and SVOCs), $k_{OH}$ is the reaction rate constant between the precursor and $OH$ (cm$^3$ molecule$^{-1}$ s$^{-1}$), $C_j^{g+p}$
is the gas + particle-phase concentration in the $j^{th}$ bin (µg m$^{-3}$), $\alpha_j$ is the mass yield of the first-generation
oxidation product of the $j^{th}$ bin (Table 2), $k_{OH,aging}$ is the reaction rate constant (cm$^3$ molecule$^{-1}$ s$^{-1}$) to
represent multi-generational aging of the oxidation products, and $\beta$ and $\gamma$ are the mass yields associated
with the production and loss terms from multi-generational aging. For the $j^{th}$ bin, the second term in
equation (2) represents the formation of oxidation products from the $j+1^{th}$ volatility bin and the third term
in equation (2) represents the loss of precursor from the $j^{th}$ bin. $\beta$ and $\gamma$ are assumed to have a value of 1
(meaning no fragmentation) but $\beta$ is zero for the last bin and $\gamma$ is zero for the first bin.

Volatility-resolved mass yields for eighteen different organic precursors for $C^*$ bins ranging from $10^{-1}$ to
$10^3$ µg m$^{-3}$ were adopted or refit based on low and high NO$_X$ parameterizations published in the literature;
organic precursors, the high and low NO$_X$ VBS mass yields, and the relevant references are listed in
Tables 2 and 3. Since there were no direct low NO$_X$ VBS parameterizations for alkanes, parameterizations
for linear, branched, and cyclic alkanes were developed using pseudo chamber data generated with the
SOM based on the low NO$_X$ parameters listed in Table 5 for $n$-dodecane, methylundecane, and
hexylcyclohexane respectively. Some of these parameterizations accounted for vapor wall losses and have
been accordingly marked in Tables 2 and 3. Each SOA precursor in the exhaust emissions was assigned a
surrogate from Table 2/3 to model SOA formation in the VBS model. When using the high NO$_X$
parameterizations, branched and cyclic alkanes were assigned surrogates based on equivalent linear
alkanes, following the work of Lim and Ziemann (2009) and Tkacik et al. (2012). A C$_X$ branched alkane
was assigned a C$_{X-2}$ linear alkane as a surrogate and a C$_X$ cyclic alkane was assigned a C$_{X+2}$ linear alkane
as a surrogate. Since we only fit alkanes up to $n$-heptadecane, we considered $n$-heptadecane as a surrogate
for alkanes C$_{17}$-C$_{22}$. The Idle-Diesel-None and Idle-Biodiesel-None experiments used the low NO$_X$
parameterizations while all the other experiments used the high NO$_X$ parameterizations. The mass transfer
(condensation/evaporation) of the VBS products to the particle phase was assumed to be kinetically-
limited in the OFR (Palm et al., 2016;Jathar et al., 2017a;Ahlberg et al., 2017); Section 2.3 describes the
mass transfer equation used to model kinetic gas/particle partitioning.

**2.2.2 Statistical Oxidation Model**
The Statistical Oxidation Model (SOM), developed by Cappa and Wilson (2012) is a semi-explicit,
parameterizable model that allows for a statistical representation of the coupled chemistry,
thermodynamic properties, and gas/particle partitioning of OA. The SOM uses a 2-dimensional carbon-
oxygen grid to track gas- and particle-phase precursors and products from precursor oxidation. Each cell
in the SOM grid represents a model organic species with a molecular weight defined by the formula
$C_xH_yO_z$. A SOM species reflects the average properties (e.g. $C^*$, reactivity) of all actual species with the
same number of carbon ($N_C$) and oxygen ($N_O$) atoms that are produced from a given precursor class (e.g.,
benzene, alkanes). In the SOM, all gas-phase species are assumed to be reactive towards OH and the OH
reaction rate constant ($k_{OH}$) is calculated using equation 3 as follows:
$$log(k_{OH}) = A_1 + A_2 \times (N_C^{A_3}) \times exp(-1 \times \frac{E_a}{8.314 \times T}) \times [1 + \frac{b_1}{\sigma\sqrt{2\pi}} exp(-\frac{1(ln(N_O+0.01)-ln(b_2)^2}{2\sigma^2})] \quad (3)$$
$\sigma \ (N_C \leq 15) = 0.0214 \times N_C + 0.5238$
$\sigma \ (N_C \geq 15) = -0.115 \times N_C + 2.695$
$b_1 = -0.2583 \times N_C + 5.8944$
$b_2 \ (N_C \leq 15) = 0.0314 \times N_C + 0.9871; \ b_2 \ (N_C > 15) = 0.25 \times N_C - 2.183$
where $A_1=15.1$, $A_2=3.94$, and $A_3=0.797$. $k_{OH}$ for a specified $N_C$ and $N_O$ is assumed to be the same for
species in all the SOM grids.

The reactions with OH lead to either functionalization or fragmentation, resulting in movement through
the carbon-oxygen grid. Six precursor-specific adjustable parameters are assigned for each SOM grid:
four parameters that define the molar yields of the four functionalized, oxidized products ($p_{O,k}$, $\Sigma p_{O,k}=1$
and hence one out of the four parameters is determined by mass balance), one parameter that determines
the probability of functionalization or fragmentation ($P_{Frag}$, $P_{Func}=1-P_{Frag}$) and one parameter that
describes the change in $C^*$ associated with the addition of one oxygen atom ($\Delta LVP$). Equation 4
represents the evolution of species in the SOM grid:
$$\frac{d[C_XO_Z]}{dt} = -k_{OH}^{X,Z}[OH][C_XO_Z] + [OH]\sum_{k=1}^{4} k_{OH}^{X,Z-k} P_{func}^{X,Z-k} p_{O,k}[C_XO_{Z-k}] +$$
$$[OH]\sum_{j=1}^{jmax}\sum_{k=0}^{kmax-Z} k_{OH}^{X+j,Z-1+k} \frac{P_{frag}^{X,Z-1+k}}{N_{fragments}^{X,Z}}[C_XO_{Z-1+k}] \quad (4)$$
where $C_XO_Z$ is the gas + particle-phase concentration of the SOM species with $X$ carbon atoms and $Z$
oxygen atoms ($\mu g \ m^{-3}$) and $N_{fragments}$ is the number of possible products from fragmentation. The
probability of fragmentation is modeled using equation 5 as a function of the O:C ratio because higher
O:C ratio compounds are expected to have a higher probability of fragmentation (Chacon-Madrid and
Donahue, 2011):
$$P_{frag} = (\frac{N_O}{N_C})^{m_{frag}} \quad (5)$$
The $C^*$ for each SOM species was calculated using equation 6 as follows:
$$log_{10}C^* = -0.337MW_{HC} + 11.56 - (N_O \times \Delta LVP) \quad (6)$$
where $MW_{HC}$ (g mole$^{-1}$) is the molecular weight of the hydrocarbon backbone (accounting only for the
carbon and hydrogen atoms).

The parameters used to model SOA formation were based on those published in Cappa et al. (2016) and
are listed in Tables 4 and 5. These parameter sets were developed by fitting the SOM predictions to
chamber measurements of SOA mass concentrations and include corrections to account for vapor wall
losses (Zhang et al., 2014). Each SOA precursor in the exhaust emissions was assigned a surrogate from
Table 4 or 5 to account for the oxidation chemistry associated with oxidation of that species. For example,
pentadecane used the parameterization developed by fitting *n*-dodecane. The difference in the initial
number of carbons and oxygens, and thus the volatility, between the surrogate compound and the
precursor compound of interest was accounted for, with consequent impact on the SOA yield. In other
words, unlike the VBS where the SOA mass yield of the SOA precursor and surrogate is identical, the
surrogate in the SOM only informed the statistical trajectory for multi-generational oxidation of a given
precursor, and the surrogate and actual compound of interest can have different SOA mass yields. The
Idle-Diesel-None and Idle-Biodiesel-None experiments used the low $NO_X$ parameters while all the other
experiments used the high $NO_X$ parameters. Similar to the VBS model, the mass transfer
(condensation/evaporation) of the SOM products to the particle phase was assumed to be kinetically-
limited in the OFR (Palm et al., 2016;Jathar et al., 2017a;Ahlberg et al., 2017) and Section 2.3 below
describes the mass transfer equation used to model kinetic gas/particle partitioning.

**2.3 Kinetic Gas/Particle Partitioning**
Palm et al. (2016), (Ahlberg et al., 2017), and Jathar et al. (2017a) have argued that the short residence
times and small condensation sinks in the OFR may not permit all low-volatility products formed from
precursor oxidation to condense onto preexisting aerosol. Hence, unlike earlier work that has assumed
equilibrium partitioning to model SOA in OFRs (Tkacik et al., 2014;Chen et al., 2013), we modeled the
kinetic gas/particle partitioning of OA using equation 7 (Zhang et al., 2014):
$$\frac{dC_i^p}{dt} = 2\pi D_i D_p N_p F_{FS}\left(C_i^g - \frac{C_i^p C_i^*}{C_{OA}}\right) \qquad (7)$$
where $C_i^p$ is the particle-phase mass concentration for the $i$[th] organic species (µg m$^{-3}$), $D_i$ is the gas-phase
diffusion coefficient of the $i$[th] organic species (m$^2$ s$^{-1}$), $D_p$ is the number mean particle diameter (m) , $N_p$ is
the total particle number concentration (m$^{-3}$), $F_{FS}$ is Fuchs-Sutugin correction for non-continuum mass
transfer, $C_i^g$ is the gas-phase mass concentration of the $i$[th] organic species (µg m$^{-3}$), $C_i^*$ is the effective
saturation concentration of the $i$[th] organic species, and $C_{OA}$ is the total OA mass concentration (µg m$^{-3}$).
The $i$[th] organic species refers to the organic compounds tracked in the VBS bins and the SOM grids. The
gas-phase diffusion coefficient was calculated for each organic species as follows:
$$D_i = D_{CO_2}\frac{MW_{CO_2}}{MW_i} \qquad (8)$$
where $D_{CO_2}$ is the gas-phase diffusion coefficient of $CO_2$ (1.38×10$^{-5}$ m$^2$ s$^{-1}$), $MW_{CO_2}$ (g mole$^{-1}$) is the
molecular weight of $CO_2$, and $MW_i$ (g mole$^{-1}$) is the molecular weight of the $i^{th}$ organic species. In the
VBS model where we do not track the molecular composition of the SOA species, we assumed all
condensing species to have a molecular weight of 200 g mole$^{-1}$. This formulation to calculate the gas-
phase diffusion coefficient under-predicted the measured gas-phase diffusion coefficients compiled by
Tang et al. (2015) by ~20%. However, doubling the gas-phase diffusion coefficient calculated in equation
8 resulted in very small change (<1%) in the OA mass predictions for a representative experiment. Hence,
we decided to use the formulation in equation (8) for the rest of this work. The Fuchs-Sutugin correction
was calculated as follows:
$$F_{FS} = \frac{0.75\alpha(1+Kn)}{Kn^2+Kn+0.283 \cdot Kn \cdot \alpha + 0.75\alpha} \qquad (9)$$
$$Kn = \frac{2\lambda_i}{D_p} \qquad (10)$$
$$\lambda_i = \frac{3D_i}{C_j} \qquad (11)$$
$$C_i = \sqrt{\frac{8N_A kT}{\pi MW_i}} \qquad (12)$$
where $Kn$ is the Knudsen number, $\alpha$ is the mass accommodation coefficient, $\lambda_i$ is the mean free path of the
$i^{th}$ organic species in air (m), $C_i$ is the root mean square speed of the gas (m s$^{-1}$), $N_A$ is Avogadro's number
(molecules mole$^{-1}$), $k$ is the Boltzmann constant (m$^2$ kg s$^{-2}$ K$^{-1}$), and $T$ is the temperature (K).

**2.4 Model Inputs**
**2.4.1 Semi-Volatile and Reactive POA**
Previous work has shown that much of combustion-related POA is semi-volatile and exists in an
equilibrium with gas-phase vapors (Robinson et al., 2007;Huffman et al., 2009;May et al., 2013c, b;May
et al., 2013a). Jathar et al. (2017a) measured emissions of POA at no OH exposure and these measured
concentrations were used to initialize the seed OA available for partitioning in the OFR and to calculate
the mass concentrations of vapors in equilibrium with the POA. The mass concentrations of the POA
vapors were determined based on the normalized, volatility-resolved distribution of primary organic
compounds estimated by May et al. (2013b) for emissions from a suite of on- and off-road diesel vehicles.
The volatility distribution of May et al. (2013b) for diesel primary organic compounds is listed in Table
6(a). For the SOM, we assumed that the primary organic compounds could be represented using a
distribution of $n$-alkanes and we refit the volatility distribution in Table 6(a) to develop a carbon-number
resolved distribution of $n$-alkanes; this distribution is listed in Table 6(b).

### 2.4.2 SOA Precursors

Jathar et al. (2017a) did not speciate the THC or SOA precursor emissions from the diesel engine and hence we have developed our own emissions profiles based on previously published literature to speciate the THC emissions. In this work, we used two different emissions profiles listed in EPA SPECIATE version 4.3 that are commonly used to speciate THC emissions from diesel engines for emissions inventories used in atmospheric modeling (EPA, 2013): Profiles #3161 (Diesel Exhaust- Farm Equipment) and #8774 (Heavy Duty Diesel Exhaust). Profile #3161 best matched the diesel engine source and diesel fuel used by Jathar et al. (2017a) and was used as the baseline emissions profile to speciate the THC emissions; we examined the sensitivity of using Profile #8774 on model predictions. We were unable to find a comprehensive emissions profile for THC emissions from the use of straight biodiesel fuel in the literature, and have relied on emissions profiles that were determined for biodiesel-diesel blends. Profile #4777 (30% Biodiesel Exhaust - Light Duty) was used as the baseline emissions profile to speciate THC emissions for experiments performed using the biodiesel fuel. All three emissions profiles (3161, 8774, and 4777) are listed in Tables S2 through S4.

Prior work in studying SOA formation has revealed that traditional speciation of THC emissions does not include emissions of high molecular-weight organic compounds, such as IVOCs, that are important SOA precursors (Jathar et al., 2014b). In Profile #3161 such compounds are partially accounted for in the 'unknown' species category (13.76% by mass of THC). Zhao et al. (2015) recently estimated that IVOCs were 60% of the THC emissions from a suite of on- and off-road diesel engines and provided a semi-explicit speciation of the IVOC emissions as a carbon-number distribution of linear, branched and cyclic alkanes. To account for these IVOC emissions, we assumed that the base case emissions profiles contained 30% IVOCs on a mass-basis (this IVOC fraction was selected since it resulted in the most optimum model-measurement comparison for OA mass; this will be discussed later in Section 3.3) and had the same chemical speciation as that proposed by Zhao et al. (2015) for an off-road engine (transportation refrigeration unit). We performed sensitivity simulations using IVOC fractions of 0% (assuming that the THC emissions contained no IVOCs), 13.76% (based on the 'unknown' category in Profile #3161), and 60% (based on the median estimate in Zhao et al. (2015)), on a mass-basis. Addition of IVOCs to the emissions profile meant that the VOC species (e.g. benzene, toluene, short alkanes) had to be renormalized to accommodate the IVOCs. Table 7 lists the renormalized baseline emissions profiles for SOA precursors used for diesel and biodiesel exhaust with 30% IVOCs along with the reaction rate constants with OH ($k_{OH}$) and surrogates (or model compound) used to model SOA formation for the VBS and SOM models. Concentrations for each species were determined by multiplying the experiment-

specific THC mass concentrations with the renormalized emissions profiles.

**2.4.3 Particle Size and Particle Number Concentrations**
For numerical simplicity, we used a monodisperse aerosol, the properties of which (number mean
diameter ($D_p$) and number concentration ($N_p$)) were initialized from the measured particle size distribution
data when modeling kinetic gas/particle partitioning. For experiments performed without the DPF+DOC,
the initial particle number concentrations and condensational sinks were high ($>3.3\times10^5$ # cm$^{-3}$ and $>0.5$
min$^{-1}$) and hence the monodisperse aerosol was initialized based on data at no photochemical exposure.
For experiments performed with the DPF+DOC where the initial particle number concentrations were
relatively low ($<1000$ # cm$^{-3}$ and $<0.003$ min$^{-1}$), photochemical aging resulted in formation and growth of
new particles and provided a substantial increase in the surface area ($>$factor of 300) available for
condensation. In these experiments, we initialized the monodisperse aerosol using an average of the data
at no photochemical exposure and after photochemical exposure (Palm et al., 2016). Averaging the data
allowed for a more realistic estimate of the condensational sink. In each simulation, the condensing SOA
mass was used to calculate the change in particle size but the number concentration was conserved. The
number mean diameter and the number concentration data - representing the initial condensational sink –
for all experiments are listed in Table 1.

New particle formation and growth was observed for most experiments at or near the highest
photochemical ages (at or $>1$ OH day), which presumably influenced the condensational sink at the
beginning of the experiment. Therefore, we performed sensitivity simulations to investigate the influence
of new particle formation on model predictions. We performed simulations with each model (VBS and
SOM) with four different initial condensational sinks. The first three simulations used measured data to
calculate the initial condensational sink inputs: (i) number mean diameter and measured number
concentration at no OH exposure (equivalent to the default for non-DPF+DOC experiments), (ii) number
mean diameter and measured number concentration at the given OH exposure, and (iii) average of (i) and
(ii) (default for DPF+DOC experiments). The fourth simulation (iv) assumed that the OFR nucleated 1
nm particles at the beginning of the experiment where the number concentration of these particles was
equal to that measured at the end of the experiment.

**2.5 Model Simulations and Model Code**
The VBS and SOM models were run separately for each photochemical exposure simulated for each
experiment listed in Table 1. In the VBS simulations, POA was tracked in one basis set while products
from each SOA precursor were tracked in separate basis sets, allowing us to distinguish between POA and
SOA. In the SOM simulations, all precursor molecules with the same surrogate (e.g., all *n*-alkanes) were
tracked in the same SOM grid. Model simulations were performed in phases to answer specific questions
and inform model inputs for later simulations:

1. To provide a general overview of the model predictions and model-measurement comparison, and
   to orient the reader to the results thereafter, we performed simulations with the VBS and SOM
   models using the base set of inputs for one of the Idle-Diesel-None experiments. Our base case
   included: Profile #3161 for VOC emissions, 30% IVOC mass fraction, kinetic gas/particle
   partitioning with a mass accommodation coefficient of 0.1, and monodisperse aerosol inputs based
   on measured data at no photochemical exposure. The partitioning- and IVOC-related choices for
   the base case are discussed in Sections 3.2 and 3.3 respectively.

2. Models used to simulate SOA production in environmental chambers and OFRs have typically
   assumed instantaneous equilibrium partitioning (e.g., Chen et al. (2013)). To examine the validity
   of assuming instantaneous equilibrium partitioning, we performed simulations with the VBS and
   SOM models using instantaneous or kinetic gas/particle partitioning for one of the Idle-Diesel-
   None and the Idle-Diesel-DPF+DOC experiments. Kinetic partitioning was modeled using three
   values of the mass accommodation coefficient ($\alpha$=0.01, 0.1, 1) to capture the uncertainty in its true
   value. To examine the influence of an increased initial condensational sink from new particle
   formation on kinetic partitioning, we performed additional simulations using four different initial
   condensational sinks (see Section 2.4.3) on one of the Idle-Diesel-None and the Idle-Diesel-
   DPF+DOC experiments.

3. Previous work has shown that combustion-related IVOCs are important precursors of SOA (e.g.,
   Jathar et al. (2014b)). To investigate the importance of IVOCs, we performed simulations with the
   VBS and SOM models at four different assumed IVOC mass fractions (0%, 13.76%, 30%, and
   60%), as discussed above when discussing the THC profiles, at all photochemical exposures and
   for all the experiments listed in Table 1. We performed additional simulations with different
   emissions profiles and SOA parameterizations on one of the Idle-Diesel-None experiments to
   investigate uncertainties linked to the composition and SOA potential of IVOCs.

4. Additional simulations were performed to examine the sensitivity of model predictions to the
   following processes: multi-generational aging, vapor wall losses, residence time distributions, and
   spatial heterogeneity in OH concentrations.

The numerical codes for the VBS were developed in Matlab while those for the SOM were developed in
IGOR (WaveMetrics Inc.). These codes will be made available on request. The simulations were
performed on an Intel i5 processor (1.7 GHz) and required ~10 s to perform a VBS simulation and ~500 s
to perform a SOM simulation at a single photochemical exposure.

**3. Results**
**3.1 General Model Results Using the Base Case**
In Figure 1, we compare predictions of OA from the VBS and SOM models using the base case to the
measurements for the Idle-Diesel-None experiment performed on June 5. Figures 1(a) and 1(b) compare
predictions to the measurements in units of µg m$^{-3}$ and g kg-fuel$^{-1}$, respectively; hereafter we present all
mass predictions in units of g kg-fuel$^{-1}$. For this experiment, the VBS/SOM models over-predicted the OA
mass by a factor of 1.9/2.2 at the lowest photochemical exposure (0.06 OH days) and a factor of 1.6/1.8 at
the next highest photochemical exposure (0.17 OH days). The over-prediction was because the models
significantly over-predicted the SOA formation at these two photochemical exposures. For higher
photochemical exposures (>0.5 OH days), both models slightly under-predicted the OA mass but
predictions were still within the measurement uncertainty. Our base case seemed to offer a mixed model-
measurement comparison for this specific experiment (i.e., over-prediction at lower photochemical ages
and a slight under-prediction at higher photochemical ages) because the 30% IVOC mass fraction used in
the base case was optimized to achieve a favorable model-measurement comparison across all
experiments at all photochemical exposures. In other words, the over-prediction in this experiment at
lower photochemical exposures was probably offset by an under-prediction at similar photochemical
exposures for some of the other experiments. It is important to note that the model performance varied
across the suite of experiments and this overall model performance is discussed in more detail in Section
3.3. The VBS and SOM models predicted that the OA at the maximum photochemical exposure was
dominated by SOA produced from VOC and IVOC oxidation (92-93%), which agreed well with the
measured composition (see Figure 1(c)). For the measurements, POA was defined as fresh OA while SOA
was defined as OA formed in addition to the POA. Furthermore, both models suggested that most of the
SOA emanated from the oxidation of IVOCs with only 7-14% resulting from the oxidation of aromatic
VOCs and less than 0.6-4% resulting from alkane VOCs smaller than a $C_{12}$. This dominance of IVOCs in
explaining the photochemically produced SOA is in line with previous OFR and chamber studies that
have modeled SOA formation from diesel exhaust (Tkacik et al., 2014;Zhao et al., 2015;Jathar et al.,
2014b).

**3.2 Kinetic Gas/Particle Partitioning**
In Figure 2, we plot predictions from the VBS and SOM models for the Idle-Diesel-None and Idle-Diesel-
DPF+DOC experiments assuming instantaneous and kinetic gas/particle partitioning. The two different
experiments were deliberately chosen to highlight the role instantaneous partitioning plays at the
extremities. We found that for the Idle-Diesel-None experiment, the use of instantaneous partitioning
roughly produced the same result as kinetic partitioning with α values of 0.1 and 1 and that all these
predictions resulted in roughly the same model-measurement comparison. The instantaneous partitioning
predictions were slightly higher than the kinetic partitioning predictions for the VBS simulations. The
kinetic partitioning simulations (except for that with an α of 0.01) produced the same result as the
instantaneous partitioning simulation most likely because the initial condensational sink was large enough
($1.12$ min$^{-1}$) in this experiment that there were no kinetic limitations to partitioning. The increase in the
condensational sink through condensation of SOA ($10$ min$^{-1}$ at the highest photochemical exposure)
tended to further reduce any differences in the predictions between the kinetic and instantaneous
partitioning simulations. However, for the Idle-Diesel-DPF+DOC experiment, the instantaneous
partitioning simulation predicted substantial OA mass at the lower photochemical exposures (0.04 and
0.12 OH days) compared to the kinetic partitioning simulations, specifically a factor of 9.8-29 larger at
0.04 OH days and a factor of 9.7-75 larger at 0.12 OH days for the VBS model and a factor of 3.9-5.8
larger at 0.04 OH days and a factor of 6.4-9.1 larger for the SOM. The instantaneous partitioning
simulations predicted a lot more SOA because all condensable products of organic precursor oxidation
were allowed to condense instantaneously (according to their respective volatilities) while the kinetic
partitioning simulations predicted little SOA production because the initial condensational sink was quite
small ($0.002$ min$^{-1}$). Predictions from the instantaneous and kinetic partitioning simulations were much
closer at the higher photochemical exposures because the SOA formed had grown the condensational sink
enough to reduce limitations to partitioning ($1$ min$^{-1}$ at the highest photochemical exposure). These results
imply that the condensation of SOA in OFRs, in some instances, could be kinetically-limited and that
instantaneous partitioning may result in models over-predicting the condensation and formation of SOA.

We make two additional observations based on the results in Figure 2. First, the initial condensational
sink for the Idle-Diesel-None experiment was large ($1.12$ min$^{-1}$) compared to condensational sinks one
would encounter in the real atmosphere. For example, 5 µg m$^{-3}$ of aerosol in a representative rural or
remote environment will have a condensational sink $<0.05$ min$^{-1}$ (Seinfeld and Pandis, 2006). Therefore,
modeling ambient applications of the OFR or OFR use with sources that use emissions control devices
will need to be even more mindful of the instantaneous partitioning assumption while predicting SOA
formation. Second, for the kinetic partitioning results, predictions from both models were relatively less
sensitive to α values between 0.1 and 1 but were dramatically lower for an α value of 0.01; more than a
factor of 2 for the Idle-Diesel-None experiment and more than an order of magnitude for the Idle-Diesel-
DPF+DOC experiment. Given the low sensitivity to α values greater than 0.1 and the reasonable model-
measurement comparison at an α value of 0.1 and 1 at least for the Idle-Diesel-None experiment, we
argue that the SOA condensation can be represented by an α value larger than 0.1 for the OFR
experiments in this work. This α value for diesel exhaust SOA was consistent with prior estimates of the α
value for biogenic SOA estimated from chamber, OFR, and aerosol heating experiments (Lee et al.,
2011;Saleh et al., 2013;Karnezi et al., 2014;Palm et al., 2016) and direct measurements of α for alkanol
SOA (Krechmer et al., 2017). However, an α of 0.1 was an order of magnitude higher than that observed
recently for toluene SOA under dry conditions (Zhang et al., 2014). Model results presented hereafter
include a kinetic treatment of gas/particle partitioning and assumed an accommodation coefficient of 0.1.

Results from model simulations performed using different initial condensational sink inputs, some of
which captured the influence of new particle formation, are plotted in Figure 3. We found that the initial
condensational sink had no influence on the OA predictions from both models for the Idle-Diesel-None
experiment, despite substantial differences in the initial condensational sink between the different cases.
This was because the amount of SOA formed (920 μg m$^{-3}$ at the highest photochemical exposure) was
sufficient to grow the condensational sink enough that the initial condensational sink did not matter. In
contrast, for both models we found large differences between the model predictions of OA for the Idle-
Diesel-DPF+DOC experiment. The use of inputs based on the measurements at no OH exposure, where
the aftertreatment system significantly reduced number concentrations (910 cm$^{-3}$) and hence the available
condensational sink (0.002 min$^{-1}$), produced much less SOA (an order of magnitude lower or more) and
poorer agreement with the measurements (see curve (i) in Figure 3(b)). Initial condensational sinks that
captured the influence of new particle formation resulted in higher model predictions but were still about
a factor of ~2 lower for the VBS simulations and a factor of ~2.7 lower for the SOM simulations when
compared against the measurements. The DPF+DOC results also suggest that calculating an initial
condensational sink using data from before and after the photochemical exposure, as done by Palm et al.
(2016), could be used as an input to model OFR data. Slight differences between the different curves for
the Idle-Diesel-None experiment and curves (ii), (iii), and (iv) for the Idle-Diesel-DPF+DOC experiment
can be attributed to the interaction of multigenerational aging and kinetic gas/particle partitioning.

**3.3 Influence of IVOCs on SOA Formation**
In Figure 4(a), we compare predictions of SOA concentrations from the SOM against measurements for
all the experiments listed in Table 1 and at all photochemical exposures. For visual clarity, we do not
present results from the VBS model as both models had nearly identical predictions with a few
exceptions; see Figure S1 where we compare VBS model predictions to SOM predictions for all
experiments at all photochemical exposures for the base case. The four panels in Figure 4(a) show model-
measurement comparisons assuming four different fractions of IVOCs: 0%, 13.76%, 30%, and 60%;
statistical metrics of fractional bias, fractional error, and $R^2$ for the comparison for both models are listed
in Table S4 (fractional bias $= \frac{1}{N}\sum_{i=1}^{n}\frac{M-O}{\frac{M+O}{2}}$, fractional error $= \frac{1}{N}\sum_{i=1}^{n}\frac{|M-O|}{\frac{M+O}{2}}$, M=predicted value,
O=observed value, N=sample size). The model-measurement comparison and the model skill was very
poor when no IVOCs were included (fractional bias = -109%, fractional error = 125%, and $R^2$ = 0.52);
this model reflects the treatment of diesel-powered sources in most traditional emissions inventories and
large-scale models. The model-measurement comparison was reasonable with 13.76% IVOCs (fractional
bias = -46%, fractional error = 101%, and $R^2$ = 0.95) but model predictions were over-predicted with 60%
IVOCs (fractional bias = 72%, fractional error = 97%, and $R^2$ = 0.99). The optimal model performance
that produced the lowest fractional bias and fractional error was realized at an IVOC mass fraction of 30%
(fractional bias= 6%, fractional error= 86%, and $R^2$ = 0.88). For predictions with an IVOC mass fraction
of 30%, 66% and 70% of the model predictions were within a factor of 1.5 and 2 of the measurements and
IVOCs on average accounted for 67%/72% (VBS/SOM) of the SOA at the highest photochemical
exposure across all experiments. Given the optimal performance, the base case used in this work assumed
a 30% IVOC fraction. These comparisons indicate that it is critical that IVOCs be included when
modeling the SOA formation from diesel exhaust and also validate the IVOC composition estimates made
by Zhao et al. (2015). We note that the model of Peng and Jimenez (2017) suggested that the organic
compounds in the OFR experiments performed by Jathar et al. (2017) may have been subjected to non-
tropospheric photolysis at 185 and 254 nm. Accounting for the photolysis of the key SOA precursors
(IVOCs and aromatics) could affect the optimal IVOC fraction identified above and hence needs to be
considered in future work.

We further investigated the IVOC species that contributed the most to SOA formation. For an IVOC
fraction of 30%, cyclic alkane IVOCs accounted for 23% of the THC emissions and on average accounted
for 61%/53% (VBS/SOM) of the SOA formation across the different experiments. We should note that
the speciation of cyclic alkane IVOCs in Zhao et al. (2015), while robust in quantifying the carbon
number, did not include any specificity in terms of the molecular structure, i.e., their methods would not
be able to distinguish between a pure $C_{10}$ cyclic alkane and a cyclohexane with a 4-carbon branch.
Further, the parameterizations to model SOA formation from cyclic alkane IVOCs for both models were
based on the behavior of particular compounds. In the VBS model when using the high $NO_X$
parameterizations, the surrogate for a cyclic alkane IVOC was determined through equivalence with a
straight alkane IVOC while in the VBS model when using the low $NO_X$ parameterizations or the SOM the
cyclic alkane IVOCs were tied to parameterizations for hexylcyclohexane. (The observed SOA yield and
derived SOM parameterization for hexylcyclohexane is actually quite similar to that for cyclododecane
for low-$NO_X$ conditions, but not for high-$NO_X$ conditions (Cappa et al., 2013)) This lack of specificity in
the speciation and the SOA parameterizations made the SOA predictions from the oxidation of cyclic
alkane IVOCs relatively uncertain. To examine the sensitivity of the model predictions to uncertainties in
the model treatment of cyclic alkane IVOCs, we performed simulations with both models for one of the
Idle-Diesel-None experiments where the cyclic alkane IVOCs were treated as branched alkane IVOCs;
results from these simulations are shown in Figure 5(a). The use of branched alkane IVOCs to model
cyclic alkane IVOCs only marginally reduced OA predictions for both the VBS and SOM models,
suggesting that the model predictions were not sensitive to the SOA parameterization used for cyclic
alkane IVOCs. Regardless, we recommend that future work focus on a more detailed speciation of cyclic
alkane IVOCs in combustion emissions as well as on chamber and OFR experiments on those speciated
compounds to improve quantification of their SOA mass yields.
As there were no direct measurements of any SOA precursors in the study of Jathar et al. (2017a), we
have used previously published emissions profiles for diesel exhaust to determine initial concentrations of
the SOA precursors. We examined the sensitivity of model predictions to two different emissions profiles
from the EPA SPECIATE (version 4.3) database: Profile #3161 (included in the base case) and Profile
#8774 that represents emissions from 'Heavy Duty Diesel Exhaust'; the speciation for both profiles is
provided in Tables S2 and S3. Both profiles only included speciation for VOC emissions and in these
simulations we assumed an IVOC mass fraction of 30%. The results captured in Figure 5(b) for one of the
Idle-Diesel-None experiments show that the choice in the emissions profile had no influence on the OA
evolution for the VBS model but had a small influence on the OA evolution for the SOM. This little to
small influence was expected given that most of the SOA was formed from IVOC, rather than VOC,
oxidation. This further demonstrates that IVOCs, not VOCs, play an important role in controlling the
SOA formation from diesel exhaust emissions and it is important that future studies work towards better
understanding the IVOC speciation.
The IVOC speciation of Zhao et al. (2015) included 37 unique species, each of which required a unique
surrogate to model the SOA formation from that species. Tracking these many IVOC species in an
atmospheric model (e.g., global climate model) may be intractable and hence, there is a need to develop
simplified parameterizations to efficiently model SOA formation from IVOCs. We note that species using
the same surrogate in the VBS model (e.g., a $C_{15}$ linear alkane, $C_{17}$ branched alkane, and $C_{13}$ cyclic alkane
are all parameterized using *n*-pentadecane when using the high $NO_X$ parameterizations) could be lumped
together to reduce the number of precursors and products tracked and that there are no penalties for a
precursor type (e.g., *n*-alkanes) to include additional precursor and product species once a SOM grid is
setup. Nonetheless, to investigate the possibility of developing a simplified parameterization, we modeled
SOA from IVOCs assuming that all the IVOCs could be modeled together as a single linear $C_{13}$, $C_{15}$, $C_{17}$
or $C_{19}$ alkane; a similar strategy was employed by (Jathar et al., 2014b) to model SOA formation from
unspeciated organic compounds in combustion emissions. Results from these simulations are shown in
Figure 5(c) for one of the Idle-Diesel-None experiments. For the VBS model, the use of a larger carbon
number alkane to model IVOC SOA produced increasingly more OA, with the $C_{19}$ alkane providing the
best comparison against measurements. For the SOM, the use of a $C_{13}$ and $C_{15}$ alkane produced good
agreement with measurements with a $C_{13}$ alkane slightly under-predicting the OA at 0.5 OH days and the
$C_{15}$ alkane slightly over-predicting the OA at lower photochemical exposures (0.06 and 0.17 OH days). It
was interesting to observe that for the SOM, in contrast to the VBS, the use of different linear alkanes
produced different OA masses at lower photochemical exposures but converged at the highest
photochemical exposure suggesting that the effective SOA mass yield in the SOM varied dynamically
with photochemical age. Differences in the VBS and SOM predictions with different alkane
parameterizations point to inherent differences in the coupled representation of multigenerational aging
and gas/particle partitioning. Results from these simulations indicate that in cases where computational
efficiency is demanded, the SOA formation from IVOCs in diesel exhaust could be modeled using a
surrogate linear alkane, possibly a $C_{19}$ linear alkane with the VBS and a $C_{13}$ or $C_{15}$ linear alkane for the
SOM.

**3.4 Elemental Composition**
The SOM tracks both the carbon and oxygen number of the oxidation products, which allowed us to
predict the O:C ratio of the OA. The O:C of the OA was calculated by combining the measured O:C of
the POA with the modeled O:C of the SOA. We compare predictions of the O:C of OA from the SOM
against measurements for all the experiments listed in Table 1 and at all photochemical exposures in
Figure 4; statistical metrics of fractional bias, fractional error, and $R^2$ for the comparison are listed in
Table S5. Model predictions for the no IVOC case, where the O:C of the OA was dominated by the O:C
of the aromatic SOA, compared well with measurements (fractional bias = -4.2%, fractional error = 28%,
and $R^2 = 0.77$). However, the poor OA mass predictions with no IVOCs suggests that the good O:C
performance was purely coincidental. The 13.76%, 30%, and 60% IVOC cases under-predicted the OA
O:C where the under-prediction appeared to increase as the IVOC influence increased; fractional bias = -
32%, fractional error = 38%, and $R^2 = 0.72$ for the 13.76% IVOC case, fractional bias = --37%, fractional
error = 42%, and $R^2 = 0.70$ for the 30% IVOC case, and fractional bias = -60%, fractional error = -62%,
and $R^2 = 0.46$ for the 60% IVOC case. A higher IVOC fraction resulted in a lower O:C ratio because the
IVOCs were primarily composed of higher carbon number species that on oxidation produced low O:C
SOA compared to SOA formed from precursors such as aromatics. On average, the 30% IVOC case
predicted an O:C ratio that was 28% lower than the measurements. For the three non-zero IVOC cases
(13.76%, 30%, and 60%), the model skill in predicting the O:C was much better for the non-DPF+DOC
experiments ($R^2 = 0.82$, 0.83, and 0.80 respectively) than for the DPF+DOC experiments ($R^2 = 0.02$, 0.02,
and 0.29 respectively). Measurements and model predictions of the OA O:C ratio from the 30% IVOC
case as a function of photochemical age are presented in Figure S2.

The under-prediction in O:C ratios was confounding when compared to earlier applications of the SOM
and in light of the reasonable model-measurement comparison found in this work in predicting OA mass.
We note that the low O:C in the 13.76%, 30%, and 60% IVOC cases stems from the dominance of
product species that have high carbon numbers and low oxygen numbers. We explored several lines of
reasoning for this under-prediction. First, Cappa et al. (2013) found good agreement between the SOM-
predicted and observed O:C for chamber experiments conducted using individual linear, branched and
cyclic $C_{12}$ alkanes. Also, general predictions of the dependence of O:C on the carbon number of the parent
hydrocarbon (cf. Fig. 2b in Cappa and Wilson (2012)) show good agreement with observations (cf. Fig.
2a in Tkacik et al. (2012)), both in terms of absolute values and shape. This suggests that uncertainties in
the SOM parameters may not be the dominant reason for the under-prediction. A possible reason for the
under-prediction then is that the compounds identified by Zhao et al. (2015) as IVOCs are structurally
different than the alkanes used to model them in this work. Second, the gas-phase chemistry in the OFR
might be inherently different than that in a chamber. For example, kinetic limitations to gas/particle
partitioning may result in gas-phase oxidation of low-volatility products having high O:C that typically
would have partitioned to the particle phase in a chamber experiment but instead are fragmented (Palm et
al., 2016). As to why the chamber-based SOM parameters then offer good model performance on OA
mass remains unclear. One way in which this issue could be addressed in the future is by developing
SOM parameters exclusively based on OFR data, as and when they become available. Third, the SOM
used here did not include surface/heterogeneous and particle-phase reactions that might influence the OA
composition and O:C ratio. When heterogeneous reactions of OA were included assuming an OH uptake
coefficient of 1 (the product distribution from the oxidation reaction was kept the same as the gas-phase
reactions), SOA production at the highest photochemical exposure for all the experiments was reduced, on
average, by 7% from fragmentation reactions within the particle phase, but the O:C ratio was only
marginally increased (average of 2%).

To understand the O:C under-prediction better, we compared model predictions of normalized gas-phase
species concentrations from the SOM to normalized gas-phase measurements made by Friedman et al.
(2017) using a chemical ionization mass spectrometer (CIMS). The CIMS detects an array of oxygenated
organic species and the high resolution of the time-of-flight mass spectrometer enables identification of
the elemental composition of each detected peak. The CIMS data were aggregated by carbon and oxygen
number to facilitate comparison with the SOM data. The comparison was performed on a normalized
basis because the CIMS did not provide absolute concentrations for every detected peak. The SOM-CIMS
comparisons for the Idle-Diesel-None and Load-Diesel-None experiments at the highest photochemical
exposure are shown in Figure 6, which highlight four findings of note. First, the CIMS measured species
larger than a carbon number of 12 that are presumably products from oxidation of higher molecular
weight organic compounds, although the possibility of dimer formation in the instrument cannot be
entirely ruled out. Nonetheless, this provides additional evidence for the presence of IVOC oxidation
products in diesel exhaust emissions. Second, the CIMS measured organic compounds with high O:C
ratios (e.g., $C_6O_6$, $C_7O_7$). This implies that the reaction chemistry in OFRs rapidly adds functional groups
to the carbon backbone, although larger, less oxidized compounds could be simultaneously functionalized
and fragmented in the CIMS leading to the appearance of highly oxidized species. Third, the SOM offered
a reasonable correlation against the CIMS measurements for both experiments across a majority of the
carbon-oxygen combinations that spanned more than four orders of magnitude. Qualitatively, this finding
validates the statistical evolution of organic compounds tracked through the generalized SOM
mechanism, although certainly some differences are evident. Finally, for the mid-carbon number species
(~$C_{10}$), the SOM seemed to produce higher fractions of species with low oxygen numbers ($O_0$ to $O_3$) but
lower fractions of species with high oxygen numbers ($O_5$ to $O_7$). This under-prediction of the high oxygen
number species might potentially explain why the SOM may be under-predicting the OA O:C ratio. The
SOM-CIMS comparison is preliminary and we intend to explore the implications of this comparison in
future work.

**3.5 Other Model Sensitivities**
We performed sensitivity analyses to examine the influence of other key processes on predictions from
both the VBS and SOM models. When examining the sensitivity to each process, all the other inputs were
kept the same as those listed in the base case. We only present sensitivity results for the Idle-Diesel-None
experiment performed on June 5, as the results for this experiment were generally representative of all
experiments (Figure 7). For completeness, we performed simulations for all the experiments at the highest
photochemical exposure since each of the processes explored below manifested the strongest response at
the highest photochemical exposure. The results from these simulations are presented as a change in the
model predictions relative to that offered by the base case.

***Multi-generational Aging.*** One of the key differences between the VBS and the SOM models is how they
represent the multi-generational aging of gas-phase products. SOA parameters for the VBS model
represent stable product distributions at the end of the chamber experiments and therefore already include
the influence of multi-generational aging reactions encountered during the chamber experiment.
Additional multi-generational aging in the VBS model, based on the scheme of Robinson et al. (2007), is
simulated as a continuous decrease in product volatility, which does not account for fragmentation
reactions and has not been constrained against experiments. The SOM framework explicitly models multi-
generational aging that includes treatment of fragmentation reactions and constrains the aging reactions
based on the chamber experiments to an extent that is determined by the length (in OH exposure space) of
the experiment. To test the influence of multi-generational aging, we performed model simulations with
aging turned off for the VBS and SOM models and plot the results in Figure 7(a). We found that aging
had a small influence (~18% reduction in OA mass) on model predictions from the VBS model, most
likely because the high SOA and OA mass concentrations resulted in a substantial fraction of the organic
species to be partitioned to the particle phase. This left very little of the organic species in the gas-phase to
participate in multi-generational aging. We calculated that less than 20% by mass of the product species
in this experiment was in the gas-phase at the OA mass concentration at the highest photochemical
exposure, implying that the SOA mass yields at these OA mass concentrations were rapidly approaching
100%. In contrast, the absence of aging resulted in a 43% decrease in the OA mass for the SOM. The
decrease was mainly because the first generation oxidation product with the highest yield (i.e., $C_xO_1$) was
too volatile to partition to the particle phase and needed to be aged further to form condensable products.
As noted earlier, the term aging is defined differently for the VBS and SOM models and the results
presented here need to take the definitional issues into account when examining the influence of aging.
Compared to the base case, no aging resulted in an average decrease of 4% and 30% in OA mass for the
VBS and SOM models respectively for all experiments at the highest photochemical exposure. These
simulations suggest that aging of the oxidation products, at least for the SOM, is as important as the
contribution of first generation products to SOA formation.

***Vapor Wall Losses.*** Prior work has highlighted the influence vapor wall losses exert on the calculation of
SOA mass yields from chamber experiments (Zhang et al., 2014;Krechmer et al., 2016). Cappa et al.
(2016), based on the chamber work of Zhang et al. (2014), recently published parameter sets for the SOM
that accounted for no vapor wall losses and two different vapor wall loss rates ($1\times10^{-4}$ and $2.5\times10^{-4}$ s$^{-1}$)
assuming an equivalent OA mass of the chamber walls of 10 mg m$^{-3}$ (the base case used the parameter
sets for a vapor wall loss rate of $2.5\times10^{-4}$ s$^{-1}$). We performed model simulations with SOM using
parameters that were either not corrected for vapor wall losses or that were corrected for vapor wall losses
using either the low ($1\times10^{-4}$) or high ($2.5\times10^{-4}$) estimates proposed by Cappa et al. (2016). The results
plotted in Figure 7(b) show that correcting for vapor wall losses significantly increased model predicted
OA mass (by 73% and 112% for the low wall loss and high wall loss cases respectively at the highest
photochemical exposure when compared to the no wall loss case) and provided the best performance for
the high estimate for vapor wall losses. Across all experiments and at the highest photochemical exposure,
accounting for vapor wall losses using the high estimate resulted in an average increase of 39% over no
accounting for vapor wall losses. These comparisons suggest that it is important to use SOA
parameterizations in which vapor wall losses in chambers have been accounted for when interpreting
SOA experiments. Furthermore, we also simulated the influence of vapor losses to the OFR walls on
model predictions. We assumed reversible uptake of vapors to the walls and used a vapor wall loss rate of
$2.5 \times10^{-3}$ s$^{-1}$ (factor of ~10 larger than that for a chamber) based on the work of Palm et al. (2016) and an
equivalent OA mass concentration of 10 mg m$^{-3}$ for the OFR walls. The results plotted in Figure 7(b)
show that the loss of vapors to the OFR walls had a small influence on model predictions: a 6.5%
decrease for this experiment and an average decrease of 11% across all experiments at the highest
photochemical exposure. Increasing the equivalent OA mass concentration for the OFR walls to 100 and
1000 mg m$^{-3}$ seemed to have no influence on model predictions. These findings imply that vapor wall
losses in the presence of sufficient seed aerosol might not be of concern for OFRs (Lambe et al., 2015).

***Residence Time Distributions.*** Model simulations performed in this work assumed that the OFR operated
as a plug flow reactor with a constant residence time. Experimental studies by Lambe et al. (2011) and
fluid dynamics simulations by Ortega et al. (2016) have shown that OFRs, particularly like the one used
in this work, exhibit heterogeneity in residence times. We performed simulations to explore the sensitivity
of varying residence times on model predictions. These simulations were performed based on a
discretized version of the residence time distribution measured by Lambe et al. (2011) for $SO_2$ that
yielded an average residence time of 100 seconds (same as that used by Jathar et al. (2017a)). The
discretized version included six parcels with volume fractions of 0.23, 0.36, 0.24, 0.11, 0.05, and 0.01
with residence times of 45, 65, 100, 200, 300, and 500 seconds respectively. Each parcel experienced the
same OH concentration but the varying residence times resulted in different OH exposures for each
parcel. The parcels were combined after photochemical exposure without repartitioning the OA between
the six parcels. Similar to the findings of Peng et al. (2015) for calculating OH exposure, our results in
Figure 7(c) show that using a residence time distribution had very little influence on the OA mass
evolution compared to use of an effective average time. Compared to the base case, the residence time
distribution resulted in an average decrease of 3% and 5% in OA mass for the VBS and SOM models
respectively for all experiments at the highest photochemical exposure.

***Spatial Heterogeneity in OH.*** In addition to the influence exerted by a distribution of residence times,
spatial heterogeneity in the gas-phase chemistry inside the OFR (e.g., from radial variation in light
intensity) could lead to spatial heterogeneity in OH concentrations and result in a distribution of OH
exposures for the sample being aged. We performed simulations to explore the sensitivity of a varying
OH exposure on model predictions. These simulations were performed where we split the sample coming
into the OFR into two parcels and treated the parcels to different OH exposures. Each experiment was
repeated for all combinations (six total) of three different parcel splits (¼-¾, ⅓-⅔, ½-½) and two different
OH exposure splits (⅓-X, ⅔-X); X was determined by conserving the total OH exposure reported by
Jathar et al. (2017a). For instance, the first simulation was performed by splitting the OFR air parcel into
¼ and ¾ fractions by volume and exposing the ¼ volume to ⅓ the OH exposure. The parcels were
combined after photochemical exposure without repartitioning the OA between the two parcels. The
results in Figure 7(d) show that the simulated spatial heterogeneity always reduced the OA mass although
the maximum reduction (13% for the VBS and 14% for the SOM models) at the highest photochemical
exposure was within the measurement uncertainty at least for the SOM. Compared to the base case, the
spatial heterogeneity in OH resulted in a maximum decrease of 13% and 15% in OA mass for the VBS
and SOM models respectively for all experiments at the highest photochemical exposure.

**4. Summary and Discussion**
Recently, Jathar et al. (2017a) reported on experiments performed using the oxidation flow reactor (OFR)
to measure the photochemical production of secondary organic aerosol (SOA) from diesel exhaust under
varying engine loads, fuel types, and aftertreatment systems. These data present an opportunity to not only
test SOA models but also use these models to interpret OFR data and determine their relevance for the
real atmosphere. In this work, we applied two different SOA model frameworks (VBS and SOM) to
simulate the photochemical production of SOA in an OFR from diesel exhaust and evaluated those model
frameworks using the data from Jathar et al. (2017a). The volatility basis set (VBS) model is a
parameterized model that allows for a volatility-based representation of OA while the statistical oxidation
model (SOM) is a semi-explicit parameterized model that uses a carbon-oxygen grid to track OA. Both
simulated the coupled chemistry, thermodynamic properties, and gas/particle partitioning of OA and
accounted for: (i) semi-volatile and reactive emissions of primary organic aerosol (POA), (ii) SOA
production from IVOCs and VOCs, (iii) multi-generational aging, and (iv) kinetic gas/particle
partitioning.

Model predictions suggest that the instantaneous gas/particle partitioning assumption may over-predict
SOA formation in OFRs when the initial condensational sinks are low and the condensation of SOA is
likely kinetically limited. Hence, SOA formation in OFRs needs to be modeled/interpreted through an
explicit treatment of kinetic gas/particle partitioning. Differences in model predictions between
instantaneous and kinetic partitioning will depend on the rate at which condensable SOA mass is
produced in the OFR (depends on the initial precursor concentrations and photochemical exposure),
residence time in the OFR, properties of the condensing species (e.g., diffusion coefficient, molecular
weight), and parameters relevant for partitioning (e.g., accommodation coefficient, seed aerosol surface
area). To explore the relative importance of instantaneous and kinetically-limited partitioning in an OFR,
we used the SOM to simulate SOA formation from diluted diesel exhaust using instantaneous and kinetic
partitioning assumptions for varying amounts of SOA formed (1-10000 $\mu g\ m^{-3}$) and initial condensational
sinks (0.001-10 $min^{-1}$). These simulations were similar to the calculations performed by Palm et al. (2017)
where they calculated timescales and losses of condensable SOA vapors to the OFR walls and sampling
lines and reaction with OH. The calculations were performed for two different initial particle sizes (10
and 100 nm) since the condensation of SOA mass would grow the initial condensational sink for the two
particles at different rates, i.e. for the same starting initial condensational sink, smaller particles would
experience quicker growth in the condensational sink compared to larger particles for the same amount of
condensing mass. The calculations were also performed for two different residence times – 2 and 4
minutes – to span the residence time range used in typical applications of the OFR. We assumed an
accommodation coefficient of 0.1. The results plotted in Figure 8 show the ratio of SOA predicted
through kinetic partitioning to that predicted through instantaneous partitioning as a function of the initial
condensational sink and the SOA formed under an instantaneous partitioning assumption. Across the four
scenarios explored (two initial particle sizes and two residence times), the SOA formation predicted under
the kinetic partitioning assumption was an order of magnitude or more lower than that predicted under the
instantaneous partitioning assumption over a large portion of the input range explored, e.g., when the
initial condensational sink was smaller than ~0.1 min$^{-1}$ and the maximum SOA formed was lower than
~100 µg m$^{-3}$ for the 10 nm simulations and lower than ~1000 µg m$^{-3}$ for the 100 nm simulations. We also
found that the SOA formation in the OFR was kinetically-limited under typical ambient conditions. The
SOA formation predicted under the kinetic partitioning assumption approached the SOA formed under the
instantaneous partitioning assumption either when the initial condensational sink was very large (>5 min$^{-}$
$^{1}$) or when a large amount of condensable SOA was produced in the OFR (>=1000 µg m$^{-3}$ for the 10 nm
particles and >>10000 µg m$^{-3}$ for the 100 nm particles). Our finding implies that ambient applications of
the OFR, where initial condensational sinks are typically smaller (~0.005-0.5 min$^{-1}$) and the maximum
SOA produced is typically less than 40 µg m$^{-3}$, will only produce a small fraction (0-30%) of the intended
SOA. Furthermore, our simulations suggested that a smaller initial particle size (i.e., 10 nm) for the same
initial condensational sink and a longer OFR residence time (i.e., 4 min) may not necessarily help produce
the intended SOA under ambient conditions. Although these simulation results need to be verified
experimentally, they do suggest that it might be challenging to operate the OFR in conditions where
instantaneous or atmospherically-relevant partitioning is applicable, further complicating the coupled
atmospheric simulation of chemistry and thermodynamics in OFRs.

Upon including IVOCs as SOA precursors, both the VBS and SOM models were able to reasonably
predict the OA mass evolution reported by Jathar et al. (2017a) across different engine loads, fuel types,
and aftertreatment systems. Model predictions suggest that 30% of the unburned hydrocarbon emissions
are likely IVOCs and that these IVOCs (regardless of the emissions profiles used to determine non-IVOC
emissions) account for most (average of 70%) of the SOA formed from diesel exhaust. These findings are
consistent with prior work from chamber experiments (Jathar et al., 2014b) and modeling studies (Zhao et
al., 2015). Simulations performed using single surrogates suggest that the complex mixture of IVOCs in
diesel exhaust could be well represented using a linear $C_{13}$ or $C_{15}$ alkane for the SOM but might need a
larger surrogate such as a $C_{19}$ alkane for the VBS model. The need for a different surrogate species to
model IVOC SOA between the SOM and VBS models most likely arises from differences in the coupled
treatment of the oxidation chemistry and gas/particle partitioning in the OFR. The use of surrogates offers
a computationally-efficient strategy to model SOA formation from IVOCs in large-scale three-
dimensional models. The SOM tracks the carbon and oxygen numbers of the oxidation products and
hence model predictions were used to calculate atomic O:C ratios for OA, which were then compared to

measurements. While the inclusion of IVOCs allowed for good model-measurement comparisons on OA mass, the SOM under-predicted the O:C ratio of OA on average by 28%, possibly highlighting the limitations in modeling the IVOCs as alkanes and/or extrapolating chamber-based parameterizations to OFR experiments. Model predictions of the gas-phase organic species compared favorably to those measured using a chemical ionization mass spectrometer (CIMS), which qualitatively validates the statistical evolution of organic compounds tracked through the generalized SOM mechanism.

As OFRs are increasingly used to study SOA formation and evolution in laboratory and field environments, there is a need to develop models that can be used to interpret OFR data. This work suggests that multi-generational aging (in case of the VBS model), residence time distributions, and spatial heterogeneity in OH concentrations produced sensitivities that were well within the measurement uncertainty and were not a concern for the model system studied. However, model predictions did appear to be more sensitive to multi-generational aging (in case of the SOM) and influence of vapor wall losses, highlighting that these processes be included in OFR models. While the conclusions from this work may be relevant for other laboratory and ambient studies, their relative importance may vary. There are several instances where the model development was insufficient and will likely be addressed in future work. For example, the model could benefit from the use of a polydisperse size distribution to treat new particle formation and growth and improve predictions of the evolution of the aerosol size distribution. The model of Peng and Jimenez (2017) suggested that the SOA precursors and their oxidation products in the Jathar et al. (2017) experiments might have been subjected to non-tropospheric photolysis. Our work did not consider the photolysis of IVOCs (or other SOA precursors), which if considered, may have implications for the IVOC findings reported here. We recommend that future studies on combustion sources significantly dilute their emissions before oxidizing them in an OFR while simultaneously accounting for photolysis reactions in models that simulate OFR chemistry. Finally, the model needs to be rigorously tested against other laboratory (e.g., Lambe et al. (2012)) and ambient (e.g., Palm et al. (2016)) OFR data.

**5 Acknowledgements**

We thank Dr. Zhe Peng and Dr. Jose Jimenez for running their oxidation flow reactor model (Peng and Jimenez, 2017) to provide $NO_X$ relevant inputs for our work. We thank the Dr. Jeffrey Pierce and Dr. John Volckens for their feedback on the written manuscript. DKF acknowledges the Beckman Young Investigator Award for funding.

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

7 Tables

*Table 1: Primary emissions of THC and POA, maximum photochemical production of SOA, maximum O:C of the OA, maximum OH exposure,*
*and size distribution data from Jathar et al. (2017a).*

| Load-Fuel-Aftertreatment Experiment | Date | THC ($\mu g\ m^{-3}$) | CO (ppbv) | NO (ppbv) | $NO_2$ (ppbv) | POA ($\mu g\ m^{-3}$) | $SOA^{\%}$ ($\mu g\ m^{-3}$) | $O:C^{\%}$ | OH Exposure[%] (molec.-hr $cm^{-3}$) | Number Mean Dia.[*] (nm) | Number Conc.[*] ($\#\ cm^{-3}$) |
|---|---|---|---|---|---|---|---|---|---|---|---|
| Idle-Diesel-None | June 3 | 1519 | 2746 | 960 | 318 | 38±15 | 209±66 | 0.23±0.01 | $2.1\times10^7$ | | $8.0\times10^5$ |
| | June 5 | 1810 | 5809 | 878 | 502 | 35±11 | 875±288 | 0.46±0.07 | $6.67\times10^7$ | 46 | $6.5\times10^5$ |
| | June 12 | 2554 | 9664 | 1870 | 1103 | 85±17 | 877±277 | 0.57±0.09 | $3.61\times10^7$ | | $3.4\times10^5$ |
| Idle-Biodiesel-None | June 4 | 1118 | 4270 | 867 | 344 | 22±12 | 999±316 | 0.52±0.07 | $9.17\times10^7$ | $46^{@}$ | $7.3\times10^{5@}$ |
| | June 8 | 2160 | 8169 | 1578 | 811 | 69±20 | 1415±468 | 0.36±0.03 | $4.72\times10^7$ | | $4.1\times10^{5@}$ |
| Load-Diesel-None | June 3 | 959 | 1558 | 4999 | 379 | 19±11 | 181±58 | 0.37±0.01 | $3.6\times10^7$ | 190 | $5.3\times10^5$ |
| | June 5 | 711 | 1400 | 6690 | 34 | 37±13 | 253±100 | 0.32±0.04 | $2.61\times10^7$ | | $4.4\times10^5$ |
| Load- Biodiesel-None | June 4 | 1634 | 1410 | 6364 | 30 | 29±18 | 645±204 | 0.38±0.05 | $2.78\times10^7$ | $190^{@}$ | $5.0\times10^{5@}$ |
| | June 8 | 518 | 2051 | 10813 | 496 | 46±22 | 284±106 | 0.30±0.04 | $1.42\times10^7$ | | $3.3\times10^{5@}$ |
| Idle-Diesel-DPF+DOC | June 9 | 2135 | 7473 | 2383 | 23 | 1.5±0.6 | 1040±335 | 0.37±0.02 | $5\times10^7$ | 52 | 910 |
| Load-Diesel-DPF+DOC | June 9 | 303 | 85 | 6157 | 4483 | 1.6±3.6 | 146±48 | 0.29±0.01 | $1.31\times10^7$ | 57 | 968 |
| Idle-Biodiesel-DPF+DOC | June 10 | 1773 | 7452 | 2213 | 182 | 2.6±1 | 787±250 | 0.44±0.04 | $5.28\times10^7$ | $52^{@}$ | $910^{@}$ |
| Load-Biodiesel-DPF+DOC | June 10 | 261 | 58 | 5475 | 4525 | 2±0.14 | 107±9 | 0.29±0.01 | $1.39\times10^7$ | $57^{@}$ | $968^{@}$ |

DPF=diesel particulate filter, DOC=diesel oxidation catalyst
[%]maximum values measured in each experiment
[*]values measured at no OH exposure
[@]No data, assumed to be similar to the equivalent diesel experiment for the model
*Table 2: SOA precursors and mass yields used in the VBS model for high NO$_X$ conditions.*

| Species | log$_{10}$C$^*$ | | | | | Reference |
|---|---|---|---|---|---|---|
| | 0.1 | 1 | 10 | 100 | 1000 | |
| toluene | 0.0000 | 0.0100 | 0.2400 | 0.4500 | 0.7000 | Hildebrandt et al. (2009) |
| benzene | 0.0392 | 0.0315 | 0.0000 | 0.8230 | 0.0957 | Ng et al. (2007)[#] |
| m-xylene | 0.0032 | 0.0106 | 0.0633 | 0.0465 | 0.0000 | Ng et al. (2007)[#] |
| p-xylene | 0.0000 | 0.0022 | 0.0764 | 0.0000 | 0.0000 | Song et al. (2007)[#] |
| o-xylene | 0.0000 | 0.0132 | 0.1140 | 0.0000 | 0.0000 | Song et al. (2007)[#] |
| naphthalene | 0.0000 | 0.1660 | 0.0000 | 0.5400 | 0.8130 | Chan et al. (2009)[#] |
| 1-methylnaphthalene | 0.0000 | 0.0170 | 0.4860 | 0.0000 | 0.0000 | Chan et al. (2009)[#] |
| 2-methylnaphthalene | 0.0000 | 0.0531 | 0.5040 | 0.0000 | 0.0000 | Chan et al. (2009)[#] |
| 1,2-dimethylnaphthalene | 0.0000 | 0.3100 | 0.0000 | 0.0000 | 0.0000 | Chan et al. (2009)[#] |
| 1-methyl-3-n-propylbenzene | 0.0000 | 0.0000 | 0.0405 | 0.0694 | 0.1140 | Odum et al. (1996)[#] |
| *n*-decane | 0.0000 | 0.0000 | 0.0110 | 0.1280 | 0.2420 | Presto et al. (2010)[&] |
| *n*-undecane | 0.0000 | 0.0040 | 0.0720 | 0.1760 | 0.1450 | Presto et al. (2010)[&] |
| *n*-dodecane | 0.0000 | 0.0140 | 0.1100 | 0.1600 | 0.0000 | Presto et al. (2010)[&] |
| *n*-tridecane | 0.0140 | 0.0590 | 0.0940 | 0.0710 | 0.0000 | Presto et al. (2010)[&] |
| *n*-tetradecane | 0.0940 | 0.3000 | 0.3500 | 0.0000 | 0.0000 | Presto et al. (2010)[&] |
| *n*-pentadecane | 0.0440 | 0.0710 | 0.4100 | 0.3000 | 0.0000 | Presto et al. (2010)[&] |
| *n*-hexadecane | 0.0530 | 0.0830 | 0.4600 | 0.2500 | 0.0000 | Presto et al. (2010)[&] |
| *n*-heptadecane | 0.0630 | 0.0890 | 0.5500 | 0.2000 | 0.0000 | Presto et al. (2010)[&] |
| *n*-octadecane | 0.0760 | 0.3195 | 0.3750 | 0.1000 | 0.0000 | Presto et al. (2010)[&] |
| *n*-nonadecane | 0.0890 | 0.5500 | 0.2000 | 0.0000 | 0.0000 | Presto et al. (2010)[&] |
| *n*-eicosane | 0.3195 | 0.3750 | 0.1000 | 0.0000 | 0.0000 | Presto et al. (2010)[&] |
| *n*-heneicosane | 0.5500 | 0.2000 | 0.0000 | 0.0000 | 0.0000 | Presto et al. (2010)[&] |
| *n*-docosane | 0.3750 | 0.1000 | 0.0000 | 0.0000 | 0.0000 | Presto et al. (2010)[&] |

[&]extrapolated from the Presto et al. (2010) data
[#]do not account for vapor wall losses
*Table 3: SOA precursors and mass yields used in the VBS model for low NO$_X$ conditions.*

| Species | C$^*$ | | | | | Reference |
|---|---|---|---|---|---|---|
| | 0.1 | 1 | 10 | 100 | 1000 | |
| toluene | 0.0000 | 0.0100 | 0.2400 | 0.7000 | 0.7000 | Hildebrandt et al. (2009) |
| benzene | 0.0000 | 0.0000 | 0.0000 | 0.3700 | 0.0000 | Ng et al. (2007)[#] |
| m-xylene | 0.0000 | 0.0000 | 0.0000 | 0.0310 | 0.0000 | Ng et al. (2007)[#] |
| p-xylene | 0.0000 | 0.0000 | 0.0000 | 0.0310 | 0.0000 | Ng et al. (2007)[#] |
| o-xylene | 0.0000 | 0.0000 | 0.0000 | 0.0310 | 0.0000 | Ng et al. (2007)[#] |
| naphthalene | 0.0000 | 0.1565 | 0.0000 | 0.1199 | 0.2708 | Chan et al. (2009)[#] |
| 1-methylnaphthalene | 0.0000 | 0.1565 | 0.0000 | 0.1199 | 0.2708 | Chan et al. (2009)[#] |
| 2-methylnaphthalene | 0.0000 | 0.1565 | 0.0000 | 0.1199 | 0.2708 | Chan et al. (2009)[#] |
| n-decane | 0.0002 | 0.0050 | 0.0013 | 0.3938 | 0.0278 | Loza et al. (2014)[&] |
| n-undecane | 0.0001 | 0.0070 | 0.0216 | 0.3321 | 0.0000 | Loza et al. (2014)[&] |
| n-dodecane | 0.0011 | 0.0080 | 0.0279 | 0.3902 | 0.0003 | Loza et al. (2014)[&] |
| n-tridecane | 0.0029 | 0.0064 | 0.0551 | 0.3231 | 0.7090 | Loza et al. (2014)[&] |
| n-tetradecane | 0.0004 | 1.2000 | 0.1777 | 0.0194 | 0.0014 | Loza et al. (2014)[&] |
| n-pentadecane | 0.0032 | 0.0124 | 0.0686 | 0.5050 | 0.0025 | Loza et al. (2014)[&] |

| | | | | | | |
|---|---|---|---|---|---|---|
| n-hexadecane | 0.0000 | 0.0572 | 0.2754 | 0.4346 | 0.1710 | Loza et al. (2014)[&] |
| n-heptadecane | 0.0399 | 0.0757 | 0.4409 | 0.3691 | 0.0000 | Loza et al. (2014)[&] |
| n-octadecane | 0.1958 | 0.0203 | 0.7077 | 0.0777 | 0.0000 | Loza et al. (2014)[&] |
| n-nonadecane | 1.0281 | 0.0000 | 0.0000 | 0.0000 | 0.0000 | Loza et al. (2014)[&] |
| n-eicosane | 0.0024 | 0.8470 | 0.2160 | 0.0000 | 0.0000 | Loza et al. (2014)[&] |
| n-heneicosane | 0.3629 | 0.6766 | 0.0250 | 0.0000 | 0.0000 | Loza et al. (2014)[&] |
| n-docosane | 0.7991 | 0.2633 | 0.0000 | 0.0000 | 0.0000 | Loza et al. (2014)[&] |
| $C_{12}$ branched alkane | 0.0077 | 0.0015 | 0.0416 | 0.2486 | 0.9179 | Loza et al. (2014)[&] |
| $C_{13}$ branched alkane | 0.0105 | 0.0007 | 0.0610 | 0.2376 | 1.2045 | Loza et al. (2014)[&] |
| $C_{14}$ branched alkane | 0.0135 | 0.0007 | 0.0819 | 0.4173 | 0.4879 | Loza et al. (2014)[&] |
| $C_{15}$ branched alkane | 0.0156 | 0.0034 | 0.1677 | 0.3553 | 0.7973 | Loza et al. (2014)[&] |
| $C_{16}$ branched alkane | 0.0075 | 0.0704 | 0.1689 | 0.5741 | 0.0000 | Loza et al. (2014)[&] |
| $C_{17}$ branched alkane | 0.0510 | 0.0000 | 0.4527 | 0.4605 | 0.0000 | Loza et al. (2014)[&] |
| $C_{18}$ branched alkane | 0.0836 | 0.0001 | 0.7962 | 0.1484 | 0.0000 | Loza et al. (2014)[&] |
| $C_{19}$ branched alkane | 0.3151 | 0.0001 | 0.7470 | 0.0000 | 0.0000 | Loza et al. (2014)[&] |
| $C_{20}$ branched alkane | 0.0198 | 0.8698 | 0.1725 | 0.0000 | 0.0000 | Loza et al. (2014)[&] |
| $C_{21}$ branched alkane | 0.3753 | 0.6837 | 0.0000 | 0.0000 | 0.0000 | Loza et al. (2014)[&] |
| $C_{22}$ branched alkane | 0.8517 | 0.2056 | 0.0000 | 0.0000 | 0.0000 | Loza et al. (2014)[&] |
| $C_{12}$ cyclic alkane | 0.0128 | 0.0302 | 0.0124 | 0.6156 | 0.0043 | Loza et al. (2014)[&] |
| $C_{13}$ cyclic alkane | 0.0297 | 0.0000 | 0.0939 | 0.4062 | 1.0776 | Loza et al. (2014)[&] |
| $C_{14}$ cyclic alkane | 0.0322 | 0.0000 | 0.1521 | 0.5341 | 0.5717 | Loza et al. (2014)[&] |
| $C_{15}$ cyclic alkane | 0.0345 | 0.0000 | 0.3430 | 0.3231 | 0.8672 | Loza et al. (2014)[&] |
| $C_{16}$ cyclic alkane | 0.0147 | 0.1426 | 0.3616 | 0.2839 | 0.6597 | Loza et al. (2014)[&] |
| $C_{17}$ cyclic alkane | 0.0574 | 0.2408 | 0.3453 | 0.4060 | 0.0000 | Loza et al. (2014)[&] |
| $C_{18}$ cyclic alkane | 0.2546 | 0.0643 | 0.6091 | 0.1431 | 0.0000 | Loza et al. (2014)[&] |
| $C_{19}$ cyclic alkane | 0.2940 | 0.2790 | 0.5010 | 0.0000 | 0.0000 | Loza et al. (2014)[&] |
| $C_{20}$ cyclic alkane | 0.3423 | 0.5700 | 0.1653 | 0.0000 | 0.0000 | Loza et al. (2014)[&] |
| $C_{21}$ cyclic alkane | 0.6100 | 0.4478 | 0.0155 | 0.0000 | 0.0000 | Loza et al. (2014)[&] |
| $C_{22}$ cyclic alkane | 0.9573 | 0.1110 | 0.0013 | 0.0000 | 0.0000 | Loza et al. (2014)[&] |

[&]produced from pseudo chamber data generated using the SOM

[#]do not account for vapor wall losses

*Table 4: SOA precursors and parameters used in the SOM (Cappa et al., 2016) for high $NO_X$ conditions.*

| Species | $m_{frag}$ | $\Delta LVP$ | $p_{O,1}$ | $p_{O,2}$ | $p_{O,3}$ | $p_{O,4}$ | Reference |
|---|---|---|---|---|---|---|---|
| *n*-dodecane | 0.0980 | 1.3900 | 0.9270 | 0.0101 | 0.0180 | 0.0445 | Loza et al. (2014) |
| methylundecane | 0.0100 | 1.2100 | 0.7419 | 0.0011 | 0.1820 | 0.0750 | Loza et al. (2014) |
| hexylcyclohexane | 0.0477 | 1.5700 | 0.7313 | 0.0381 | 0.2101 | 0.0205 | Loza et al. (2014) |
| toluene | 0.2220 | 1.2400 | 0.0029 | 0.0010 | 0.0010 | 1.0100 | Zhang et al. (2014) |
| benzene | 0.5350 | 1.7000 | 0.0792 | 0.0010 | 0.9190 | 0.0010 | Ng et al. (2007) |
| *m*-xylene | 0.0100 | 1.6800 | 0.9360 | 0.0010 | 0.0021 | 0.0609 | Ng et al. (2007) |
| naphthalene | 0.1210 | 1.3100 | 0.6440 | 0.0010 | 0.0460 | 0.3080 | Chan et al. (2009) |

*Table 5: SOA precursors and parameters used in the SOM (Cappa et al., 2016) for low $NO_X$ conditions.*

| Species | $m_{frag}$ | $\Delta LVP$ | $p_{O,1}$ | $p_{O,2}$ | $p_{O,3}$ | $p_{O,4}$ | Reference |
|---|---|---|---|---|---|---|---|
| *n*-dodecane | 2.0000 | 1.8300 | 0.9990 | 0.0010 | 0.0010 | 0.0010 | Loza et al. (2014) |

| methylundecane | 2.8200 | 1.9100 | 0.9980 | 0.0010 | 0.0010 | 0.0010 | Loza et al. (2014) |
| hexylcyclohexane | 5.0000 | 2.0500 | 0.8160 | 0.1810 | 0.0019 | 0.0010 | Loza et al. (2014) |
| toluene | 1.3100 | 1.7700 | 0.1850 | 0.0010 | 0.0019 | 0.8120 | Zhang et al. (2014) |
| benzene | 0.0807 | 1.9700 | 0.6370 | 0.0010 | 0.0021 | 0.3600 | Ng et al. (2007) |
| *m*-xylene | 1.0800 | 2.0500 | 0.1020 | 0.0010 | 0.8780 | 0.0190 | Ng et al. (2007) |
| naphthalene | 0.1890 | 1.8700 | 0.3520 | 0.0543 | 0.5330 | 0.0609 | Chan et al. (2009) |

*Table 6: (a) Volatility- and (b) carbon-number resolved distributions used to determine mass concentrations of POC in the VBS and SOM models respectively. The volatility distributions are from (May et al., 2013b)..*

| $C^*$ ($\mu g\ m^{-3}$) | $10^{-2}$ | $10^{-1}$ | $10^1$ | $10^2$ | $10^3$ | $10^4$ | $10^5$ | $10^6$ |
|---|---|---|---|---|---|---|---|---|
| $f_i$ | 0.03 | 0.25 | 0.37 | 0.23 | 0.06 | 0.03 | 0.01 | 0.01 |

| Carbon No. | <16 | 16 | 17 | 18 | 19 | 20 | 21 | 22 | 23 | 24 | 25 | 26 | >26 |
|---|---|---|---|---|---|---|---|---|---|---|---|---|---|
| $f_i$ | 0.003 | 0.000 | 0.058 | 0.043 | 0.055 | 0.094 | 0.146 | 0.181 | 0.178 | 0.137 | 0.078 | 0.026 | 0.001 |

*Table 7: Reaction rate constants ($k_{OH}$), mass fractions, and VBS and SOM surrogates for SOA precursors in diesel and biodiesel emissions. $k_{OH}$ values are from Atkinson and Arey (2003) when available or the EPI Suite version 4.11 (EPA, 2017).*

| Species | Carbon Number | $k_{OH}$ ($cm^3$ molecules$^{-1}$ s$^{-1}$) | Mass Percentage of THC | | VBS Surrogate | SOM Surrogate |
|---|---|---|---|---|---|---|
| | | | Diesel | Biodiesel | | |
| ethylbenzene | 8 | $7.0\times10^{-12}$ | 0.2516 | 0.0826 | toluene | toluene |
| indan | 9 | $1.9\times10^{-11}$ | 0.1542 | NA | naphthalene | naphthalene |
| butylbenzene | 10 | $4.5\times10^{-12}$ | 0.0081 | 0.4720 | *m*-xylene | *m*-xylene |
| diethylbenzene | 10 | $8.11\times10^{-12}$ | 0.0731 | NA | *m*-xylene | *m*-xylene |
| isopropyltoluene | 10 | $8.54\times10^{-12}$ | NA | 0.3599 | toluene | toluene |
| *m*-xylene | 8 | $2.31\times10^{-11}$ | 0.4951 | 0.3717 | *m*-xylene | *m*-xylene |
| *o*-xylene | 8 | $1.36\times10^{-11}$ | 0.2760 | 0.3953 | *o*-xylene | *m*-xylene |
| *p*-xylene | 8 | $1.43\times10^{-11}$ | 0.0812 | NA | *p*-xylene | *m*-xylene |
| *n*-decane | 10 | $1.1\times10^{-11}$ | 0.4302 | 1.7050 | *n*-decane | *n*-decane |
| *n*-undecane | 11 | $1.23\times10^{-11}$ | 0.2110 | 1.9410 | *n*-undecane | *n*-dodecane |
| toluene | 7 | $5.63\times10^{-12}$ | 1.1932 | 1.6401 | toluene | toluene |
| *n*-tridecane | 13 | $1.68\times10^{-11}$ | NA | 0.6136 | *n*-tridecane | *n*-dodecane |
| benzaldehyde | 7 | $1.2\times10^{-11}$ | 0.5682 | NA | benzene | benzene |
| benzene | 6 | $1.22\times10^{-12}$ | 1.6234 | 1.5988 | benzene | benzene |
| $C_{10}$ aromatics | 10 | $2.3\times10^{-11}$ | 0.0649 | NA | *m*-xylene | *m*-xylene |
| $C_9$ aromatics | 9 | $2.31\times10^{-11}$ | 0.4058 | NA | *m*-xylene | *m*-xylene |
| 1,2,3-trimethylbenzene | 9 | $3.27\times10^{-11}$ | 0.0974 | NA | *m*-xylene | *m*-xylene |
| 1,2,4-trimethylbenzene | 9 | $3.25\times10^{-11}$ | 0.4302 | 0.4720 | *m*-xylene | *m*-xylene |

| | | | | | | |
|---|---|---|---|---|---|---|
| 1,2-diethylbenzene | 10 | $8.11\times10^{-12}$ | 0.0731 | NA | toluene | toluene |
| 1,3,5-trimethylbenzene | 9 | $5.67\times10^{-11}$ | NA | 0.1888 | $m$-xylene | $m$-xylene |
| 1,2-dimethyl-4-ethylbenzene | 10 | $1.69\times10^{-11}$ | NA | 0.176 | $m$-xylene | $m$-xylene |
| 1,3-dimethyl-2-ethylbenzene | 10 | $1.76\times10^{-11}$ | NA | 0.3304 | $m$-xylene | $m$-xylene |
| 1,4-dimethyl-2-ethylbenzene | 10 | $1.69\times10^{-11}$ | NA | 0.4366 | $m$-xylene | $m$-xylene |
| 1-(1,1-dimethylethyl)-3,5-dimethylbenzene | 12 | $3.01\times10^{-11}$ | NA | 0.3717 | $m$-xylene | $m$-xylene |
| 1-methyl-2-ethylbenzene | 9 | $7.44\times10^{-12}$ | 0.1136 | 0.3835 | toluene | toluene |
| 1-methyl-3-ethylbenzene | 9 | $1.39\times10^{-11}$ | 0.2029 | 0.7198 | toluene | toluene |
| 1-methyl-2-tert-butylbenzene | 11 | $6.74\times10^{-12}$ | NA | 0.4307 | toluene | toluene |
| 1-tert-butyl-4-ethylbenzene | 12 | $7.42\times10^{-12}$ | NA | 0.1947 | $m$-xylene | $m$-xylene |
| 2-methyl-butyl-benzene | 11 | $1.02\times10^{-11}$ | NA | 1.1032 | $m$-xylene | $m$-xylene |
| 3,3-dimethyloctane | 10 | $7.21\times10^{-12}$ | NA | 0.3068 | $n$-decane | methylundecane |
| 3-ethyloctane | 10 | $1.18\times10^{-11}$ | NA | 0.1888 | $n$-decane | methylundecane |
| 3-methylnonane | 10 | $1.14\times10^{-11}$ | NA | 0.2655 | $n$-decane | methylundecane |
| $C_{12}$ branched alkane | 12 | $1.82\times10^{-11}$ | 1.1335 | 1.1335 | $n$-decane | methylundecane |
| $C_{13}$ branched alkane | 13 | $1.68\times10^{-11}$ | 0.8111 | 0.8111 | $n$-undecane | methylundecane |
| $C_{14}$ branched alkane | 14 | $1.39\times10^{-11}$ | 0.5257 | 0.5257 | $n$-dodecane | methylundecane |
| $C_{15}$ branched alkane | 15 | $1.82\times10^{-11}$ | 0.4692 | 0.4692 | $n$-tridecane | methylundecane |
| $C_{16}$ branched alkane | 16 | $1.96\times10^{-11}$ | 0.4935 | 0.4935 | $n$-tetradecane | methylundecane |
| $C_{17}$ branched alkane | 17 | $2.1\times10^{-11}$ | 0.2198 | 0.2198 | $n$-pentadecane | methylundecane |
| $C_{18}$ branched alkane | 18 | $2.24\times10^{-11}$ | 0.2863 | 0.2863 | $n$-hexadecane | methylundecane |
| $C_{19}$ branched alkane | 19 | $2.38\times10^{-11}$ | 0.1716 | 0.1716 | $n$-heptadecane | methylundecane |
| $C_{20}$ branched alkane | 20 | $2.52\times10^{-11}$ | 0.0969 | 0.0969 | $n$-octadecane | methylundecane |
| $C_{21}$ branched alkane | 21 | $2.67\times10^{-11}$ | 0.0639 | 0.0639 | $n$-nonadecane | methylundecane |
| $C_{22}$ branched alkane | 22 | $2.81\times10^{-11}$ | 0.0604 | 0.0604 | $n$-eicosane | methylundecane |
| $C_{12}$ cyclic alkane | 12 | $1.82\times10^{-11}$ | 4.3427 | 4.3427 | $n$-tetradecane | hexylcyclohexane |
| $C_{13}$ cyclic alkane | 13 | $1.68\times10^{-11}$ | 4.4265 | 4.4265 | $n$-pentadecane | hexylcyclohexane |
| $C_{14}$ cyclic alkane | 14 | $1.39\times10^{-11}$ | 3.1480 | 3.1480 | $n$-hexadecane | hexylcyclohexane |
| $C_{15}$ cyclic alkane | 15 | $1.82\times10^{-11}$ | 2.8599 | 2.8599 | $n$- | hexylcyclohexane |

| | | | | | heptadecane | |
|---|---|---|---|---|---|---|
| $C_{16}$ cyclic alkane | 16 | $1.96\times10^{-11}$ | 2.1848 | 2.1848 | *n*-octatadecane | hexylcyclohexane |
| $C_{17}$ cyclic alkane | 17 | $2.1\times10^{-11}$ | 1.8546 | 1.8546 | *n*-nonadecane | hexylcyclohexane |
| $C_{18}$ cyclic alkane | 18 | $2.24\times10^{-11}$ | 1.6900 | 1.6900 | *n*-eicosane | hexylcyclohexane |
| $C_{19}$ cyclic alkane | 19 | $2.38\times10^{-11}$ | 1.0570 | 1.0570 | *n*-heneicosane | hexylcyclohexane |
| $C_{20}$ cyclic alkane | 20 | $2.52\times10^{-11}$ | 0.5900 | 0.5900 | *n*-docosane | hexylcyclohexane |
| $C_{21}$ cyclic alkane | 21 | $2.67\times10^{-11}$ | 0.3736 | 0.3736 | *n*-tricosane | hexylcyclohexane |
| $C_{22}$ cyclic alkane | 22 | $2.81\times10^{-11}$ | 0.3141 | 0.3141 | *n*-tricosane | hexylcyclohexane |
| dodecane | 12 | $1.82\times10^{-11}$ | 0.5830 | 0.5830 | *n*-dodecane | *n*-dodecane |
| tridecane | 13 | $1.68\times10^{-11}$ | 0.5465 | 0.5465 | *n*-tridecane | *n*-dodecane |
| tetradecane | 14 | $1.39\times10^{-11}$ | 0.3649 | 0.3649 | *n*-tetradecane | *n*-dodecane |
| pentadecane | 15 | $1.82\times10^{-11}$ | 0.3063 | 0.3063 | *n*-pentadecane | *n*-dodecane |
| hexadecane | 16 | $1.96\times10^{-11}$ | 0.2281 | 0.2281 | *n*-hexadecane | *n*-dodecane |
| heptadecane | 17 | $2.1\times10^{-11}$ | 0.1655 | 0.1655 | *n*-heptadecane | *n*-dodecane |
| octadecane | 18 | $2.24\times10^{-11}$ | 0.1481 | 0.1481 | *n*-octatadecane | *n*-dodecane |
| nonadecane | 19 | $2.38\times10^{-11}$ | 0.0726 | 0.0726 | *n*-nonadecane | *n*-dodecane |
| eicosane | 20 | $2.52\times10^{-11}$ | 0.0365 | 0.0365 | *n*-eicosane | *n*-dodecane |
| heneicosane | 21 | $2.67\times10^{-11}$ | 0.0222 | 0.0222 | *n*-heneicosane | *n*-dodecane |
| docosane | 22 | $2.81\times10^{-11}$ | 0.0143 | 0.0143 | *n*-docosane | *n*-dodecane |
| pristane | 19 | $2.44\times10^{-11}$ | 0.1434 | 0.1434 | *n*-nonadecane | methylundecane |
| phytane | 20 | $2.61\times10^{-11}$ | 0.0799 | 0.0799 | *n*-eicosane | methylundecane |
| naphthalene | 10 | $2.3\times10^{-11}$ | 0.1038 | 0.1038 | naphthalene | naphthalene |
| phenanthrene | 14 | $1.3\times10^{-11}$ | 0.0117 | 0.0117 | naphthalene | naphthalene |

169

## 8 Figures

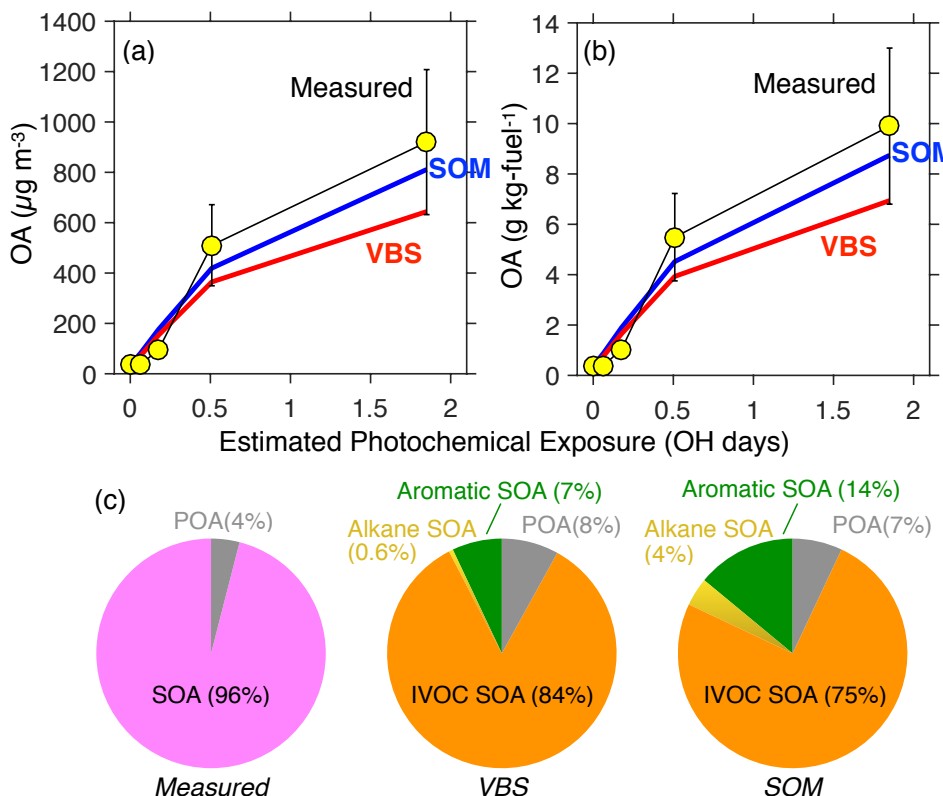

*Figure 1: VBS and SOM model predictions of OA compared to measurements from the experiment*

*performed on June 5 (Idle-Diesel-None) as a function of photochemical age. Inputs for both models have*

*been specified in the text. Panel (a) has comparisons in µg m$^{-3}$ and panel (b) has comparisons in g kg-*

*fuel$^{-1}$. Panel (c) shows the modeled and measured OA composition at the highest photochemical exposure.*

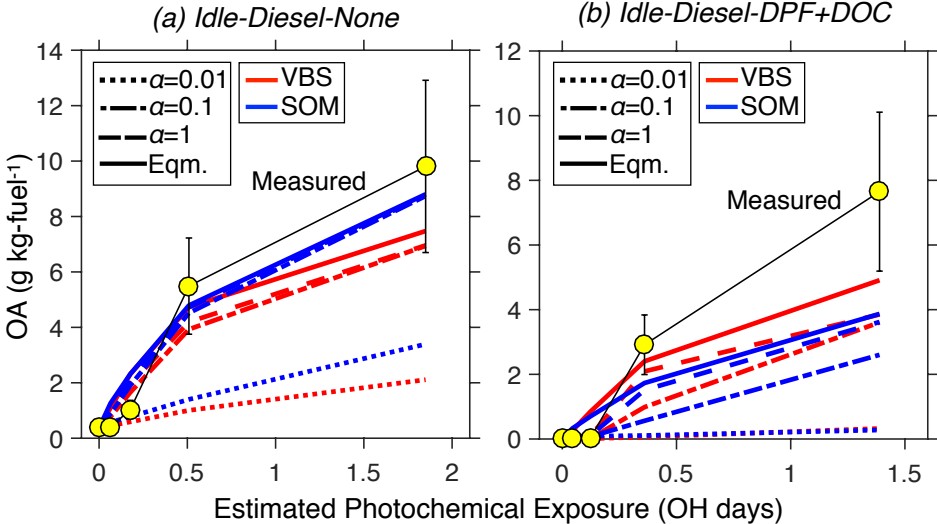

*Figure 2: VBS model predictions of OA compared to measurements from the experiment performed on*

*June 5 (Idle-Diesel-None) as a function of photochemical age assuming instantaneous equilibrium*

*partitioning and kinetic gas/particle partitioning run at three accommodation coefficients, α = 1 (dash-*

*dot) ,0.1 (dash) and 0.01 (solid).*

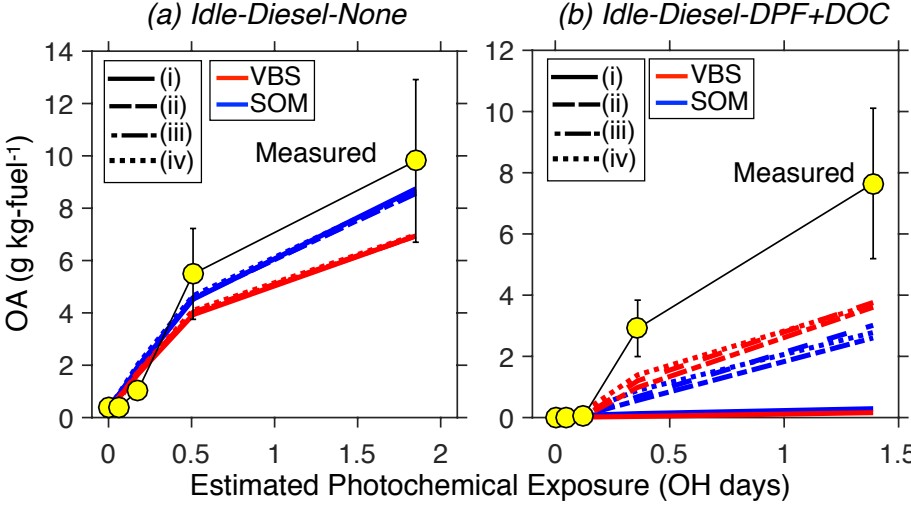

*Figure 3: VBS and SOM model predictions of OA compared to measurements from the experiments*
*performed on (a) June 5 (Idle-Diesel-None) and (b) June 11 (Idle-Diesel-DPF+DOC) as a function of*
*photochemical age for four different particle size distribution inputs: (i) number mean diameter and*
*measured number concentration at no OH exposure (solid), (ii) number mean diameter and measured*
*number concentration at the given OH exposure (dash), (iii) average of (i) and (ii) (dash-dot), and (iv)*
*nucleation of 1 nm particles (dot).*

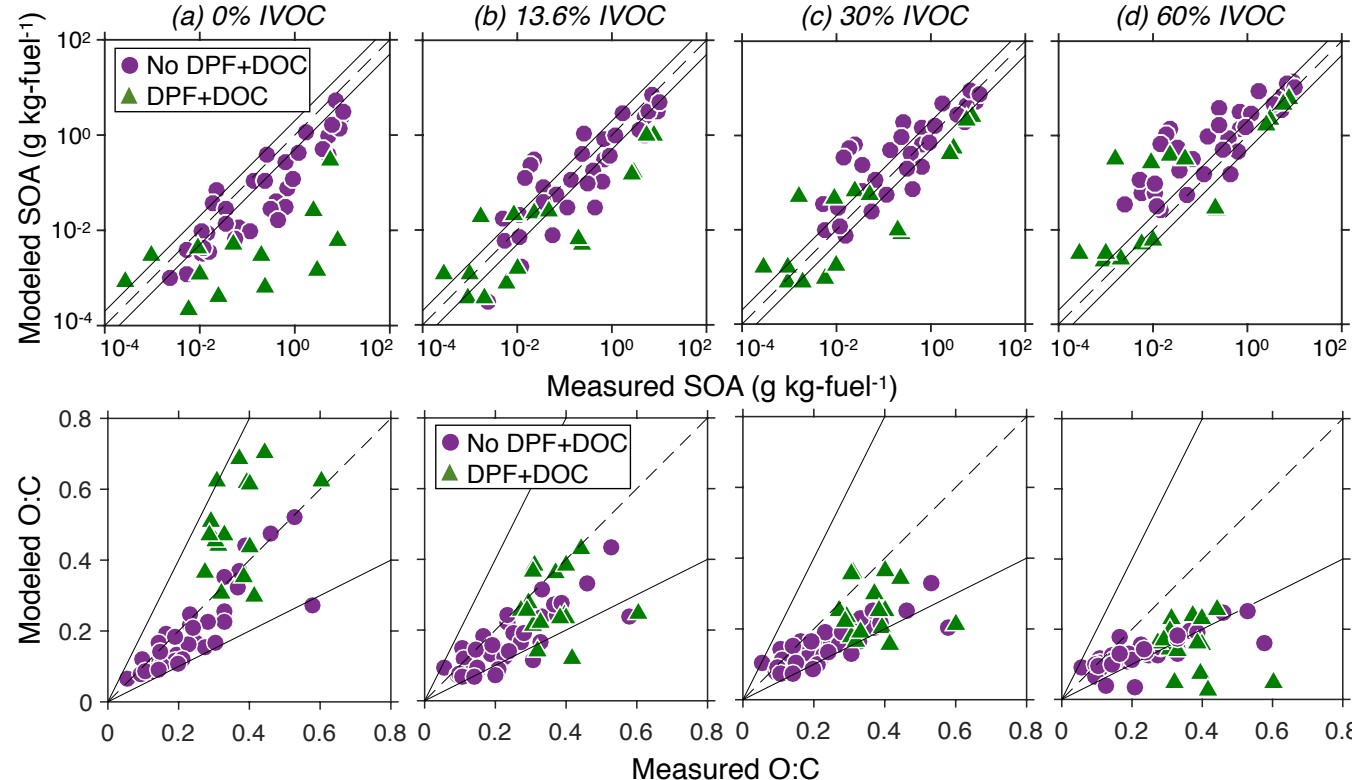

*Figure 4: Scatter plot comparing SOM predictions of OA mass and O:C to measurements from all*
*experiments at all photochemical ages at four different IVOC mass fractions: (a) 0%, (b) 13.76%, (c)*
*30%, and (d) 60%.*

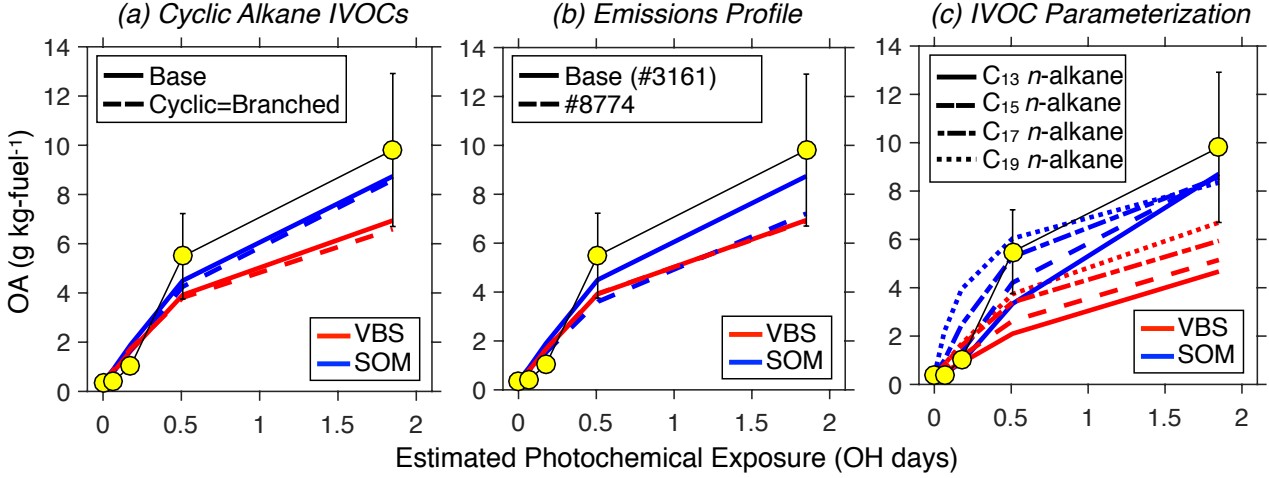

*Figure 5: VBS and SOM predictions of OA compared to measurements from the experiment performed on June 5 (Idle-Diesel-None) as a function of photochemical age. Panel (a) examines uncertainty in model treatment of cyclic alkanes, panel (b) examines uncertainty in the VOC emissions profile, and (c) explores suitability of using a single surrogate linear alkane to model SOA formation from all IVOCs.*

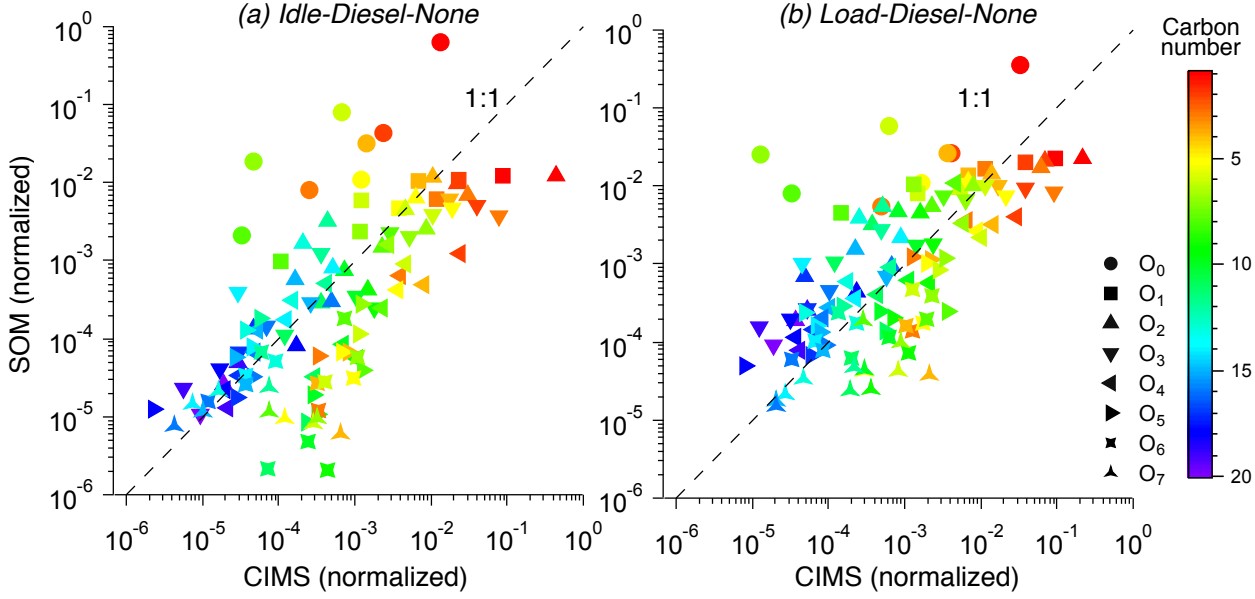

*Figure 6: Normalized gas-phase concentration predictions from the SOM model for the Idle-Diesel-None and Load-Diesel-None experiments performed on June 5 and compared to normalized gas-phase concentrations measured by the CIMS.*

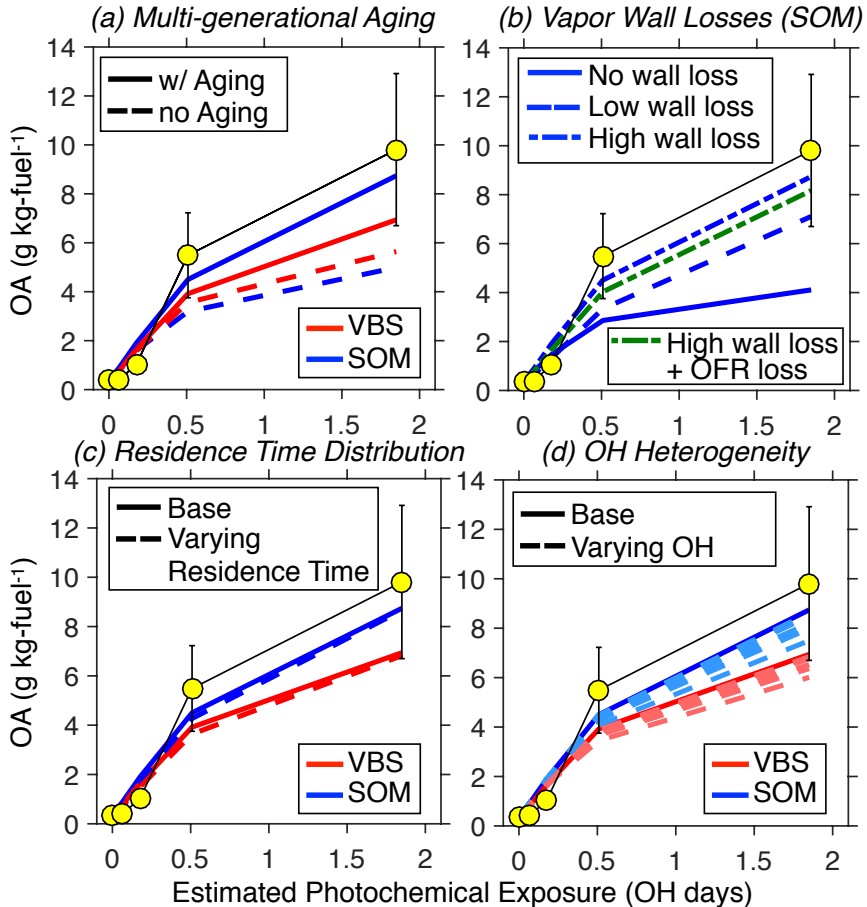

209

*Figure 7: VBS and SOM predictions of OA compared to measurements from the experiment performed on*
*June 5 (Idle-Diesel-None) as a function of photochemical age. Panels (a), (b), (c), and (d) examine the*
*influence of multi-generational aging, vapor wall losses, residence time distribution, and spatial*
*heterogeneity in OH concentrations respectively. The dashed lines in panel (d) are deliberately lighter in*
*color than the solid line to help differentiate the Base result from the sensitivity results.*


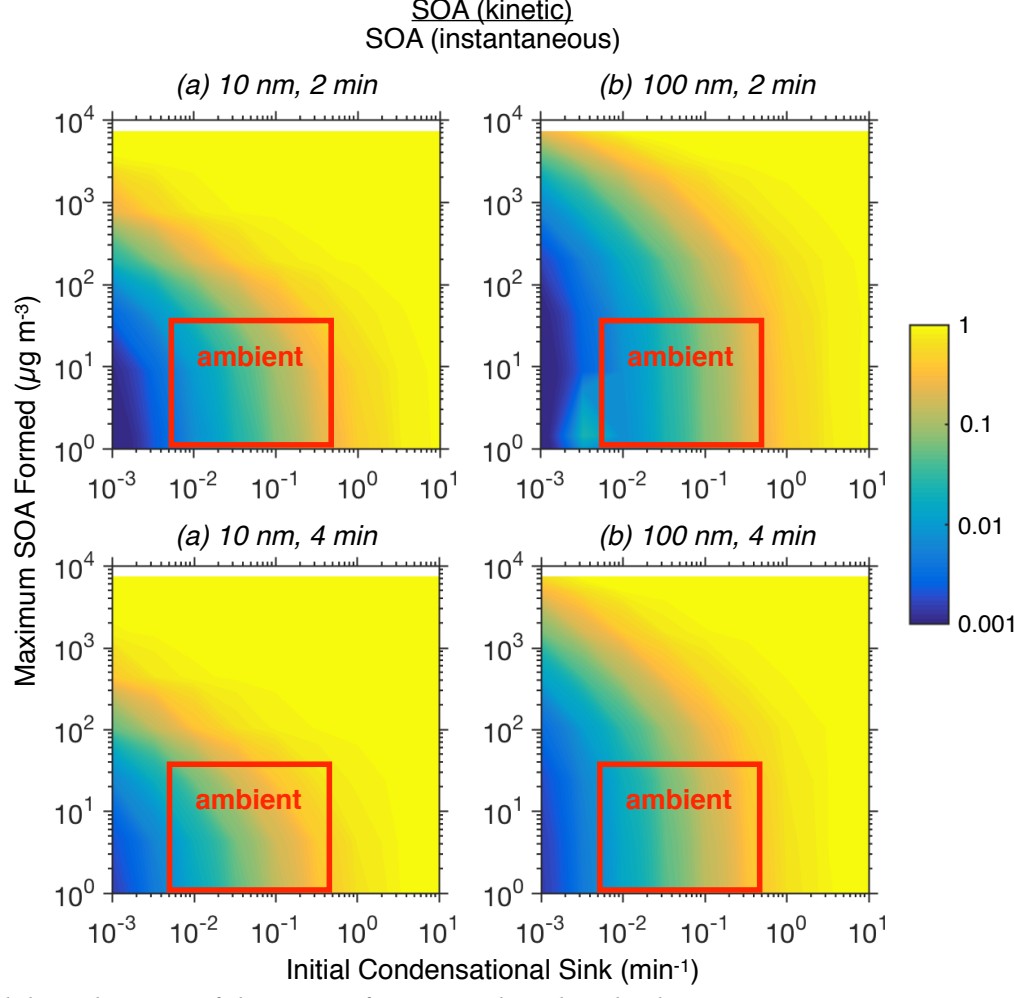

*Figure 8: Model predictions of the ratio of SOA produced under kinetic partitioning assumptions to the*
*SOA produced under instantaneous partitioning assumptions as a function of the initial condensational*
*sink and the SOA formed under instantaneous partitioning. Panels are for calculations performed at two*
*different particles sizes: 10 and 100 nm and two different residence times: 2 and 4 min.*

