# Peer review of "Modeling the Formation and Composition of Secondary Organic"

_Atmospheric Chemistry and Physics, 2017_

## Referee Comment (RC1) · Anonymous Referee #3 · 24 Jan 2018

The authors present VBS and SOM modeling of SOA formed from oxidation of a range of diesel exhaust conditions in an OFR. They compare the models with measurements of both aerosol mass and atomic O:C. In general, the models more readily agree with the mass of SOA formed, while some disagreement was observed for O:C. Possible reasons for this disagreement are discussed. A brief but favorable comparison with gas phase CIMS measurements was also presented. Sensitivity of the models with regard to a variety of parameters was investigated. In general, this analysis is very thorough and clearly presented. It represents an important step towards understanding

the chemistry and physics that occurs in OFR experiments, and how such experiments can be used to learn about atmospheric chemistry and physics. I would recommend this paper for publication in ACP after addressing these minor comments:

General Comments:

Pg. 4 Ln. 149: Somewhere in Sect. 2.1, can you state the OFR residence time used in these experiments? It must be given in Jathar 2017a, but it would be useful to quote it here directly for context when you discuss specific condensational sink lifetimes below.

Pg. 7 Ln. 271: How well does this method of estimation of the diffusion coefficient work? Can you provide any references where this method has been tested and used before? If not, please state your reasoning for using it, or better yet include a simple figure comparing the values of D estimated using this equation with independent measurements/calculations of D for several typical species, to illustrate that it works well enough. (It might also be the case that your results are insensitive to variations in D).

Pg. 10 Ln. 357: Your numbers 1 through 3 here are mostly repeating information you have already given the reader in the prior sections (number 4 is mostly new text). It's OK to leave this text if you want to summarize and emphasize the paper organization, but you could consolidate/remove the duplicate text to streamline your Methods section.

Pg. 18 Ln. 691-696: I think it would really enhance this section if you could expand upon your discussion of Fig. 8 and related text. For instance, you've calculated for a 100s residence time, but what does it look like for the residence times of 2-4 min that have been used in most of the ambient studies, e.g. Palm et al. 2016 that you cite previously? How does this compare to the discussion in Sect. 3.3 of Palm et al. 2016, which is similar to the analysis presented here? They have suggested that the use of seed aerosol (to increase condensational sink) and the use of longer residence times could avoid such issues; can you use your results to provide support for such conclusions, or better yet provide quantitative guidance for future OFR studies (e.g.

use residence times greater than X min or a minimum seed surface area)? Such suggestions would provide a great reference for the growing OFR user community.

Technical Comments:

Pg. 5 Ln. 168: Please define 'THC'.

Table 1: Are some/all of the values reported in this table already reported in the Jathar et al 2017a reference, or are they new data presented in this work? If the former, you should cite that paper in the table caption.

Pg. 5 Ln. 192: It's confusing that you're using 'VOC' to represent all of the organic gases including SVOCs and IVOCs. I suggest you refer to them as 'organic gases' instead of 'VOCs' in such instances, so you keep the nomenclature clean.

Fig. 1c: Please label the yellow pie slices in the VBS and SOM pies.

[Figure]

---

## Referee Comment (RC2) · Anonymous Referee #2 · 8 Feb 2018

Eluri et al. modeled their OFR experiments with diesel engine exhausts by VBS and SOM. Their work is detailed. They also considered a number of sensitivity studies and meticulously articulated how the studies were performed. The authors obtained a lot of interesting results to the OFR community, including both those confirming previous observations (e.g., the importance of S/IVOC in SOA formation and the possible kinetic limitation for OA growth in OFR) and new ones (e.g., minor impacts of RTD and OH spatial heterogeneity on SOA formation). This study fits well in the scope of ACP and is by and large of good quality. The paper will be acceptable into ACP after the issues

below are addressed.

Major issues:

- Peng and Jimenez (2017) have published a characterization of whether the chemistry in OFRs with high initial NO is similar or very different from that in the atmosphere. The results on Figure 4 of that paper can be used to approximately determine whether the cases studied here fall in the good / risky /bad regions and whether the chemistry is predominantly in a low or high NO regime. This is important information that should be included in the paper.

- High-NOx parametrizations were used in this study, while Peng and Jimenez (2017) found that high-NOx conditions are very unlikely to be realized in OFR by initial NO injection only. (This can be verified with the previous suggestion) Low-NOx SOA yields are often significantly higher than high-NOx ones and SOA formed under low-NOx conditions tends to have higher O:C (the results on elemental ratio in this study might be an evidence for this). The paper acknowledges that OA is 65% higher in a sensitivity test when low-NOx yields were used. The validity of the modeling in this study is thus questionable, and the agreement in e.g. Figure 1 appears to be for the wrong reasons. The authors should show their simulation results for experiments using low-NOx parametrizations and correct their conclusions if needed.

Specific comments:

Line 209 and elsewhere: Ahlberg et al. (2017) did systematic experiments and VBS modeling to show that SOA growth in OFR is kinetically limited at low OA mass loading. This paper also needs to be cited for discussions on the kinetic limitation of SOA growth.

Line 292: The POA VBS has more bins than the SOA ones shown in Table 2. How are the extra POA bins treated? Do they participate in aging?

Line 360: The reason why alpha=0.1 is assumed is not clear to me until I arrive at the

end of Section 3.2. A brief explanation or a reference to Section 3.2 is needed.

Line 386: Why do curves in Panels a and b of Figure 1 look slightly different? E.g. the measurement point at highest age is located between the VBS and SOM predictions in Figure 1a while it overlaps with the VBS one in Figure 1b.

Line 388: Why does an overprediction of x3 occur at low ages? This should be discussed further.

Line 431: Palm et al. (2016) reported that a low value of the accommodation coefficient («1) was inconsistent with their ambient OFR SOA formation experiments. A more direct measurement of the accommodation coefficient of SOA (compared to the more indirect measurements on the publications cited here) has been recently published by Krechmer et al. (2017), with measured values in the range 0.5-1. Both of these results support the use of higher values for accommodation coefficient and thus support the conclusions here.

Line 443: In Figure 3b, Curves (iv) appear to be higher than Curves (ii). However, the condensational sink of (ii) is always that at the end of the growth while that of (iv) starts from 1 nm particles (identical number concentration). The average of the latter is obviously lower than that of the former and hence less OA will condense in the case of (iv) in theory. Why does Figure 3b show the opposite?

Line 564: Figure 6 is supposed to convey important information. But it is impossible for me to tell data points for similar carbon numbers, so that I cannot verify relevant claims made in Sections 3.4. I suggest using different markers/patterns instead of point size to distinguish different carbon numbers.

Technical corrections:

Table 1: the 3rd column from the right is OH exposure, not OH.

Table 1: the initial NO and NO2 concentrations should be given here, given their importance.

Tables 2 and 3: many references in these tables are not listed on the reference list.

Equation 3: a parameter, As, seems to be explained as "A3" in the text. Please maintain the consistency.

Line 330: although spelled out in a table caption, "DPF+DOC" still needs to be spelled out here.

Line 455: a fractional bias < -100% does not look appropriate, although I understand what the authors mean. I suggest comparing them in log scale or using ratio instead of bias.

Figure 7d: the colors of the dashed lines are not correct.

Figure 8a: the exponent of "10ˆ1" in the x-axis is covered.

References:

Ahlberg, E., et al: Secondary organic aerosol from VOC mixtures in an oxidation flow reactor, Atmos. Environ., 161, 210–220, doi:10.1016/j.atmosenv.2017.05.005, 2017.

Krechmer, J. E. et al. Direct measurements of gas/particle partitioning and mass accommodation coefficients in environmental chambers. Environ. Sci. Technol., 51, 11867–11875, doi:10.1021/acs.est.7b02144, 2017.

Palm, B. B., et al.: In situ secondary organic aerosol formation from ambient pine forest air using an oxidation flow reactor, Atmos. Chem. Phys., 16, 2943-2970, https://doi.org/10.5194/acp-16-2943-2016, 2016.

Peng, Z., et al.: Non-OH chemistry in oxidation flow reactors for the study of atmospheric chemistry systematically examined by modeling, Atmos. Chem. Phys., 16(7), 4283–4305, doi:10.5194/acp-16-4283-2016, 2016.

Peng, Z. and Jimenez, J. L.: Modeling of the chemistry in oxidation flow reactors with high initial NO, Atmos. Chem. Phys., 17(19), 11991–12010, doi:10.5194/acp-

17-11991-2017, 2017.

---

## Author Response (AR1)

We thank both reviewers for their comments. We have revised the manuscript based on their comments and queries and provided a point-by-point response below. Reviewer comments are in regular black, our response is in blue, and the additions/updated text from the manuscript are in red.

**Reviewer 1**

1. Pg. 4 Ln. 149: Somewhere in Sect. 2.1, can you state the OFR residence time used in these experiments? It must be given in Jathar 2017a, but it would be useful to quote it here directly for context when you discuss specific condensational sink lifetimes below.
The OFR residence time used in Jathar et al. (2017) was 100 seconds. We have mentioned the residence time in the revised text: "The OFR had a residence time of 100 s.".

2. Pg. 7 Ln. 271: How well does this method of estimation of the diffusion coefficient work? Can you provide any references where this method has been tested and used before? If not, please state your reasoning for using it, or better yet include a simple figure comparing the values of D estimated using this equation with independent measurements/calculations of D for several typical species, to illustrate that it works well enough. (It might also be the case that your results are insensitive to variations in D).

[Figure]

The method to estimate the gas-phase diffusion coefficient was previously used by Zhang et al. (2014) to model SOA formation from toluene photo-oxidation using the statistical oxidation model (SOM). We have added the Zhang et al. (2014) reference before we introduce the equation in the revised text. The diffusion coefficients calculated using equation 7 in the text were evaluated against diffusion coefficients compiled by Tang et al. (2015) for a suite of organic compounds. The comparison is shown in the figure to the side. The equation used in this work under-predicted the diffusion coefficient by ~30% for species with a value lower than $10^{-5}$ $m^2$ $s^{-1}$. We examined the sensitivity in the model predictions to the use of a higher diffusion coefficient. Increasing the diffusion coefficient by a factor of two did not have any impact on the model predictions of OA for the Idle-Diesel-None experiment performed on June 5. The following text was added to the methods section after describing the equations for calculating the gas-phase diffusion coefficient: "This formulation to calculate the gas-phase diffusion coefficient under-predicted the measured gas-phase diffusion coefficients compiled by Tang et al. (2015) by ~20%. However, doubling the gas-phase diffusion coefficient calculated in equation 8 resulted in very small change (<1%) in the OA mass predictions for a representative experiment. Hence, we decided to use the formulation in equation (8) for the rest of the work."

3. Pg. 10 Ln. 357: Your numbers 1 through 3 here are mostly repeating information you have already given the reader in the prior sections (number 4 is mostly new text). It's OK to leave this text if you want to summarize and emphasize the paper organization, but you could consolidate/remove the duplicate text to streamline your Methods section.

We agree with the reviewer that the sections preceding section 2.5 already describe the various simulations performed in this work. However, section 2.5 provides an overview of all the simulations for the reader to review/jump back to when reading through the results.

4. Pg. 18 Ln. 691-696: I think it would really enhance this section if you could expand upon your discussion of Fig. 8 and related text. For instance, you've calculated for a 100s residence time, but what does it look like for the residence times of 2-4 min that have been used in most of the ambient studies, e.g. Palm et al. 2016 that you cite previously? How does this compare to the discussion in Sect. 3.3 of Palm et al. 2016, which is similar to the analysis presented here? They have suggested that the use of seed aerosol (to increase condensational sink) and the use of longer residence times could avoid such issues; can you use your results to provide support for such conclusions, or better yet provide quantitative guidance for future OFR studies (e.g. use residence times greater than X min or a minimum seed surface area)? Such suggestions would provide a great reference for the growing OFR user community.

Based on the reviewer's suggestion, we performed simulations with an OFR residence time of 2 and 4 mins for both the 10 and 100 nm initial particle sizes. In our original text, we had commented on the influence of the initial condensational sink. We found that the initial condensational sink had to be very large ($>5$ min$^{-1}$) to produce the same SOA as that under instantaneous partitioning conditions and hence would be very hard to achieve in ambient applications of the OFR. If the initial condensational sink were smaller, the use of a smaller particle size distribution could help but would still be insufficient in ambient applications of the OFR. In the revised text, simulations with the two different residence times suggested that the residence times did not have a significant impact on our conclusions. Based on the simulation results, we recommend that the OFR be operated with very high initial condensational sinks ($>1$ min$^{-1}$) in case the SOA formed is less than 100 µg m$^{-3}$. If larger amounts of SOA are expected to be formed, it might be possible to operate the OFR with lower initial condensational sinks. We also state that these results will need to be experimentally verified. We have expanded on the original text as follows: "To explore the relative importance of instantaneous and kinetically-limited partitioning in an OFR, we used the SOM to simulate SOA formation from diluted diesel exhaust using instantaneous and kinetic partitioning assumptions for varying amounts of SOA formed (1-10000 µg m$^{-3}$) and initial condensational sinks (0.001-10 min$^{-1}$). These simulations were similar to the calculations performed by Palm et al. (2017) where they calculated timescales and losses of condensable SOA vapors to the OFR walls and sampling lines and reaction with OH. The calculations were performed for two different initial particle sizes (10 and 100 nm) since the condensation of SOA mass would grow the initial condensational sink for the two particles at different rates, i.e. for the same starting initial condensational sink, smaller particles would experience quicker growth in the condensational sink compared to larger particles for the same amount of condensing mass. The calculations were also performed for two different residence times – 2 and 4 minutes – to span the residence time range used in typical applications of the OFR. We assumed an accommodation coefficient of 0.1. The results plotted in Figure 8 show the ratio of SOA predicted through kinetic partitioning to that predicted through instantaneous partitioning as a function of the initial condensational sink and the SOA formed under an instantaneous partitioning assumption. Across the four scenarios explored (two initial particle sizes and two residence times), the SOA formation predicted under the kinetic partitioning assumption was an order of magnitude or more lower than that predicted under the instantaneous partitioning assumption over a large portion of the input range explored, e.g., when the initial condensational sink was smaller than ~0.1 min$^{-1}$ and the maximum SOA formed was lower than ~100 µg m$^{-3}$ for the 10 nm simulations and lower than ~1000 µg m$^{-3}$ for the 100 nm simulations. We also found that the SOA formation in the OFR was kinetically-limited under typical ambient conditions. The SOA formation predicted under the kinetic partitioning assumption approached the SOA formed under the instantaneous partitioning assumption either when the initial condensational sink was very large (>5 min$^{-1}$) or when a large amount of condensable SOA was produced in the OFR (>=1000 µg m$^{-3}$ for the 10 nm particles and >>10000 µg m$^{-3}$ for the 100 nm particles). Our finding implies that ambient applications of the OFR, where initial condensational sinks are typically smaller (~0.005-0.5 min$^{-1}$) and the maximum SOA produced is typically less than 40 µg m$^{-3}$, will only produce a small fraction (0-30%) of the intended SOA. Furthermore, our simulations suggested that a smaller initial particle size (i.e., 10 nm) for the same initial condensational sink and a longer OFR residence time (i.e., 4 min) may not necessarily help produce the intended SOA under ambient conditions. Although these simulation results need to be verified experimentally, they do suggest that it might be challenging to operate the OFR in conditions where instantaneous or atmospherically-relevant partitioning is applicable, further complicating the coupled atmospheric simulation of chemistry and thermodynamics in OFRs.".

Technical Comments:
Pg. 5 Ln. 168: Please define 'THC'.
This has been corrected.

Table 1: Are some/all of the values reported in this table already reported in the Jathar et al 2017a reference, or are they new data presented in this work? If the former, you should cite that paper in the table caption.
All values in Table 1, except for the information on the particle size distribution, have been reported in Jathar et al. (2017a). We have added the citation to the figure caption.

Pg. 5 Ln. 192: It's confusing that you're using 'VOC' to represent all of the organic gases including SVOCs and IVOCs. I suggest you refer to them as 'organic gases' instead of 'VOCs' in such instances, so you keep the nomenclature clean.
We have revised the text in the entire manuscript to refer to VOCs, SVOCs, and IVOCs together as organic precursors.

Fig. 1c: Please label the yellow pie slices in the VBS and SOM pies.
We apologize for the omissions. The yellow pie refers to the SOA formed from alkanes. The label has been added to Figure 1(c).

**Reviewer 2**

1. Peng and Jimenez (2017) have published a characterization of whether the chemistry in OFRs with high initial NO is similar or very different from that in the atmosphere. The results on Figure 4 of that paper can be used to approximately determine whether the cases studied here fall in the good / risky /bad regions and whether the chemistry is predominantly in a low or high NO regime. This is important information that should be included in the paper.

We thank the reviewer for this important comment. Based on the information in Table 2, the experiments of Jathar et al. (2017a) were most appropriately represented by the ML and MM cases although some of the experiments were beyond the MM case. The figure below shows the data from this work overlaid on Figure 4 from Peng and Jimenez (2017). This comparison suggested that we use the high NO$_X$ parameterizations to model SOA formation for most of the experiments in this work.

[Figure]

To improve our precision, we contacted Zhe Peng (first author of the Peng and Jimenez (2007) paper) and Zhe helped us run their model to determine low versus high NO conditions for all our experiments. For each experiment, the model was run with initial values of the external OH reactivity contributed by CO and THC, NO and NO$_2$ concentrations, the photon flux at each photochemical exposure, and the relative humidity. Model results suggested that the OFR ran in a high NO mode at all photochemical exposures when the engine was run at load conditions or with an aftertreatment device in place. However, the model predicted that the OFR ran in a low NO mode at most of the photochemical exposures when the engine was run at idle conditions and without an aftertreatment device (i.e., Idle-Diesel-None and Idle-Biodiesel-None). These findings from the Peng and Jimenez (2017) model were used to inform the choice between low and high NO$_X$ parameterizations to model SOA formation. The following text was added to Section 2.1 to briefly describe the problem and results from the Peng and Jimenez (2017) model: "Although the diesel exhaust was diluted with clean air to produce atmospherically-relevant concentrations of POA, the initial THC, CO, and NO$_X$ concentrations in the OFR were still quite high. Peng and Jimenez (2017) using a detailed gas-phase model argued that the high external OH reactivity might lead to non-OH chemistry in the OFR and NO could quickly be consumed in the OFR leading to low NO conditions for SOA formation. Peng and Jimenez (2017) quantified the potential influence of NO on the oxidation chemistry by calculating the ratio of the reactive flux of the peroxy radicals with NO to the reactive flux of the peroxy radicals with HO$_2$ (r$_{RO2+HO2}$/r$_{RO2+NO)}$. A ratio greater than 1 was considered as "high NO" while a ratio less than 1 was considered "low NO". For the relative humidity, photon flux, initial NO, and external OH reactivity values in Jathar et al. (2017a), the model of Peng and Jimenez (2017) predicted that the OFR ran in a high NO mode at all photochemical exposures when the engine was run at load conditions or with an aftertreatment device in place. However, the model predicted that the OFR ran in a low NO mode especially at the high photochemical exposures when the engine was run at idle conditions and without an aftertreatment device (i.e., Idle-Diesel-None and Idle-Biodiesel-None). The r$_{RO2+HO2}$/r$_{RO2+NO}$ ratio and low versus high NO mode for each experiment and photochemical exposure is listed in Table S1. Based on these results, we accordingly used the low and high NO$_X$ parameterizations to perform the model simulations.".

The low NO$_X$ SOA parameterizations for the VBS and SOM models were included in Tables 3 and 5 respectively. All figures and the corresponding discussion in the results section were revised based on the use of the new parameterizations. For the full set of changes, please refer to the marked Word document included along with the response.

2. High-NOx parametrizations were used in this study, while Peng and Jimenez (2017) found that high-NOx conditions are very unlikely to be realized in OFR by initial NO injection only. (This can be verified with the previous suggestion) Low-NOx SOA yields are often significantly higher than high-NOx ones and SOA formed under low-NOx conditions tends to have higher O:C (the results on elemental ratio in this study might be an evidence for this). The paper acknowledges that OA is 65% higher in a sensitivity test when low-NOx yields were used. The validity of the modeling in this study is thus questionable, and the agreement in e.g. Figure 1 appears to be for the wrong reasons. The authors should show their simulation results for experiments using low- NOx parametrizations and correct their conclusions if needed.

As mentioned in the previous comment, we used the Peng and Jimenez (2017) model to determine if we needed the low versus high NO$_X$ parameterization to model the SOA formation across the experimental matrix and at different photochemical exposures. The Peng and Jimenez (2017) model recommended the use of low NO$_X$ parameterizations for the Idle-Diesel/Biodiesel-None experiments and the use of high NO$_X$ parameterizations for all the remaining experiments. The use of an experiment/photochemical exposure-specific SOA parameterization resulted in three key changes to the manuscript. First, this resulted in a modest change in the IVOC fraction that allowed for the most optimum model-measurement comparison across the entire experimental matrix: 30% in the revised work versus 40% in the original work. Second, it resulted in a better O:C comparison at the optimum IVOC fraction compared to the original work and produced slight differences in the SOM versus CIMS comparison performed in Figure 6. Third, the use of a low NO$_X$ parameterization to model SOA resulted in a substantial increase in the SOA formed in the Idle-Diesel-None experiment, which was used as the representative experiment to discuss results in Figures 1, 2, 3, 5, 6, and 7. To keep the discussion in these figures centered around understanding the influence of the chemistry, thermodynamic, or OFR process, we updated the IVOC fraction used in the base case to ensure a good model-measurement comparison for this particular experiment. A lot of small changes have been made in the manuscript to reflect this update so please refer to the marked Word document included along with the response.

3. Line 209 and elsewhere: Ahlberg et al. (2017) did systematic experiments and VBS modeling to show that SOA growth in OFR is kinetically limited at low OA mass loading. This paper also needs to be cited for discussions on the kinetic limitation of SOA growth.

We thank the reviewer for pointing this out. We have folded the Ahlberg et al. (2017) study and cited it along with Lambe et al. (2015), Palm et al. (2016), and Jathar et al. (2017a).

4. Line 292: The POA VBS has more bins than the SOA ones shown in Table 2. How are the extra POA bins treated? Do they participate in aging?

The VBS in our numerical model is initialized with C* bins ranging for $10^{-2}$ to $10^6$ µg m$^{-3}$ to ensure we accommodate the full range of POA volatilities. For first-generation SOA, the C* bins other than those listed in Table 2 and 3 are zeroed out. Multigenerational aging for both POA and SOA is modeled using the scheme of Robinson et al. (2007) where mass in each C* bin is allowed to react with OH and yield a product with a $C^*$ that is an order of magnitude lower than the direct precursor, to a lower limit $C*$ of $10^{-2}$ µg m$^{-3}$.

5. Line 360: The reason why alpha=0.1 is assumed is not clear to me until I arrive at the end of Section 3.2. A brief explanation or a reference to Section 3.2 is needed.
We have added a note here referring to the relevant sections: "The choices for the base case are discussed in Sections 3.2 and 3.3.".

6. Line 386: Why do curves in Panels a and b of Figure 1 look slightly different? E.g. the measurement point at highest age is located between the VBS and SOM predictions in Figure 1a while it overlaps with the VBS one in Figure 1b.
Thank you for pointing that out. The measurements in Figure 1(b) were not being calculated correctly. This has been corrected.

7. Line 388: Why does an overprediction of x3 occur at low ages? This should be discussed further.
The over-prediction in Figure 1 that simulates the OA evolution using the base case is mostly a result of choosing a 30% IVOC fraction. The 30% IVOC fraction was chosen for the base case to produce the best global model-measurement comparison across all photochemical exposures and all experiments. The model-measurement comparison, visualized in Figure 4(c) (top panel), exhibits a fractional bias of 6% suggesting that the use of a 30% IVOC fraction did not over- or under-predict the OA mass with photochemical exposure. It just so happens that the 30% IVOC fraction for the Idle-Diesel-None experiment over-predicted the OA mass at lower photochemical exposures but produced a reasonable comparison at higher photochemical exposures. In contrast, the use of the 30% IVOC fraction resulted in the opposite effect for the Idle-Diesel-DPF+DOC experiment shown in Figure 2(b) where the base case model (dashed lines) produced reasonable agreement at the lower photochemical exposures but under-predicted the OA mass at higher photochemical exposures. We have added the following text to address this comment: "Our base case seemed to offer a mixed model-measurement comparison for this specific experiment (i.e., over-prediction at lower photochemical ages and a slight under-prediction at higher photochemical ages) because the 30% IVOC mass fraction used in the base case was optimized to achieve a favorable model-measurement comparison across all experiments at all photochemical exposures. In other words, the over-prediction in this experiment at lower photochemical exposures was probably offset by an under-prediction at similar photochemical exposures for some of the other experiments. It is important to note that the model performance varied across the suite of experiments and this overall model performance is discussed in more detail in Section 3.3.".

8. Line 431: Palm et al. (2016) reported that a low value of the accommodation coefficient («1) was inconsistent with their ambient OFR SOA formation experiments. A more direct measurement of the accommodation coefficient of SOA (compared to the more indirect measurements on the publications cited here) has been recently published by Krechmer et al.

(2017), with measured values in the range 0.5-1. Both of these results support the use of higher values for accommodation coefficient and thus support the conclusions here.

The studies of Palm et al. (2016) and Krechmer et al. (2017) are cited as part of the mass accommodation discussion: "This α value for diesel exhaust SOA was consistent with prior estimates of the α value for biogenic SOA estimated from chamber, OFR, and aerosol heating experiments (Lee et al., 2011;Saleh et al., 2013;Karnezi et al., 2014;Palm et al., 2016) and direct measurements of α for alkanol SOA (Krechmer et al., 2017). However, an α of 0.1 was an order of magnitude higher than that observed recently for toluene SOA under dry conditions (Zhang et al., 2014). Model results presented hereafter include a kinetic treatment of gas/particle partitioning and assumed an accommodation coefficient of 0.1.".

9. Line 443: In Figure 3b, Curves (iv) appear to be higher than Curves (ii). However, the condensational sink of (ii) is always that at the end of the growth while that of (iv) starts from 1 nm particles (identical number concentration). The average of the latter is obviously lower than that of the former and hence less OA will condense in the case of (iv) in theory. Why does Figure 3b show the opposite?

The OA mass predictions for curves (ii), (iii), and (iv) are quite similar and the small differences between them can be attributed to the coupled interaction of multi-generational aging and kinetic gas/particle partitioning. The following text has been added: "Slight differences between the different curves for the Idle-Diesel-None experiment and curves (ii), (iii), and (iv) for the Idle-Diesel-DPF+DOC experiment can be attributed to the interaction of multigenerational aging and kinetic gas/particle partitioning.".

10. Line 564: Figure 6 is supposed to convey important information. But it is impossible for me to tell data points for similar carbon numbers, so that I cannot verify relevant claims made in Sections 3.4. I suggest using different markers/patterns instead of point size to distinguish different carbon numbers.

The figures (see below) have been revised to show carbon numbers in color and oxygen numbers in symbols.

[Figure]

11. Table 1: the 3rd column from the right is OH exposure, not OH.

This has been corrected.

12. Table 1: the initial NO and NO2 concentrations should be given here, given their importance.
The NO and $NO_2$ concentrations along with the CO concentrations have been included in Table 1.

13. Tables 2 and 3: many references in these tables are not listed on the reference list.
The missing references have been added to the references list.

14. Equation 3: a parameter, As, seems to be explained as "A3" in the text. Please maintain the consistency.
The variable $A_s$ in equation 3 should have been $A_3$. This has been corrected.

15. Line 330: although spelled out in a table caption, "DPF+DOC" still needs to be spelled out here.
The following detail about the DPF+DOC was added in Section 2.1: "The engine was run at two different loads (idle and 50% load) with two different fuels (diesel and biodiesel), and with and without an aftertreatment system. The aftertreatment system included a diesel oxidation catalyst (DOC) to oxidize CO and THC and a diesel particle filter (DPF) to trap fine particles.".

16. Line 455: a fractional bias < -100% does not look appropriate, although I understand what the authors mean. I suggest comparing them in log scale or using ratio instead of bias.
Fractional bias and fractional error are statistical metrics very commonly used to evaluate model predictions in the air quality and atmospheric chemistry community. The fractional bias can vary from -200% to +200% while the fractional error varies between 0% to 200%. An under-prediction of a factor of 2 translates to a fractional bias of -67%, an under-prediction of a factor of 5 translates to a fractional bias of -133%, and an under-prediction of a factor of 10 translates to a fractional bias of -163%. We have added the formulae for fractional bias and fractional error where we discuss these statistical metrics for the first time: "statistical metrics of fractional bias, fractional error, and $R^2$ for the comparison for both models are listed in Table S4 (fractional bias $= \frac{1}{N}\sum_{i=1}^{n}\frac{M-O}{\frac{M+O}{2}}$, fractional error $= \frac{1}{N}\sum_{i=1}^{n}\frac{|M-O|}{\frac{M+O}{2}}$, M=predicted value, O=observed value, N=sample size)".

17. Figure 7d: the colors of the dashed lines are not correct.
The colors of the dashed lines are deliberately lighter to separate them from the base case simulation result. This fact has been added to the caption: "The dashed lines in panel (d) are deliberately lighter in color than the solid line to help differentiate the Base result from the sensitivity results.".

18. Figure 8a: the exponent of "10^1" in the x-axis is covered.
This has been fixed in the new figure.

**Modeling the Formation and Composition of Secondary Organic Aerosol from Diesel Exhaust Using Parameterized and Semi-Explicit Chemistry and Thermodynamic Models**

Sailaja Eluri[1], Christopher D. Cappa[2], Beth Friedman[3], Delphine K. Farmer[3], and Shantanu H. Jathar[1]

[1] Department of Mechanical Engineering, Colorado State University, Fort Collins, CO, USA, 80523

[2] Department of Civil and Environmental Engineering, University of California Davis, Davis, CA, USA, 95616

[3] Department of Chemistry, Colorado State University, Fort Collins, CO, USA, 80523

Correspondence to: Shantanu H. Jathar (shantanu.jathar@colostate.edu)

**Abstract**

Laboratory-based studies have shown that combustion sources emit volatile organic compounds that can be photo-oxidized in the atmosphere to form secondary organic aerosol (SOA). In some cases, this SOA can exceed direct emissions of primary organic aerosol (POA). Jathar et al. (2017) recently reported on experiments that used an oxidation flow reactor (OFR) to measure the photochemical production of SOA from a diesel engine operated at two different engine loads (idle, load), two fuel types (diesel, biodiesel) and two aftertreatment configurations (with and without an oxidation catalyst and particle filter). In this work, we used two different SOA models, the volatility basis set (VBS) model and the statistical oxidation model (SOM), to simulate the formation and composition of SOA for those experiments. Leveraging recent laboratory-based parameterizations, both frameworks accounted for a semi-volatile and reactive POA; SOA production from semi-volatile, intermediate-volatility, and volatile organic compounds (SVOC, IVOC and VOC); $NO_X$-dependent parameterizations, multigenerational gas-phase chemistry; and kinetic gas/particle partitioning. Both frameworks demonstrated that for model predictions of SOA mass to agree with measurements across all engine load-fuel-aftertreatment combinations, it was necessary to model the kinetically-limited gas-particle partitioning in OFRs as well as account for SOA formation from IVOCs, which were on average found to account for 70% of the model-predicted SOA. Accounting for IVOCs however resulted in an average under-prediction of 28% for OA atomic O:C ratios. Model predictions of the gas-phase organic compounds (resolved in carbon and oxygen space) from the SOM compared favorably to gas-phase measurements from a Chemical Ionization Mass Spectrometer (CIMS), substantiating the semi-explicit chemistry captured by the SOM. Model-measurement comparisons were improved on using vapor wall-loss corrected SOA parameterizations. As OFRs are increasingly used to study SOA formation and evolution in laboratory and field environments, models such as those developed in this work can be used to interpret the OFR data.
* * *
**Comment [Office1]:** Sailaja: this needs to come from a comment made later.

**Comment [Office2]:** Sailaja: this needs to come from a comment made later.

[revised manuscript text omitted]

[#]generated pseudo parameters from SOM

The mass fractions now need to be included for an IVOC mass fraction of 30%.

[Figure]

[Figure]

[Figure]

Page 7: [6] Deleted      Microsoft Office User      4/26/18 1:06:00 PM

*three*

*three*

*three*

---

## Author Response (AR2)

We thank reviewer 2 for his/her comment. We have revised the manuscript and added the following text at four different places in the main manuscript. We have also attached the marked manuscript in pdf format to locate the changes.

[revised manuscript text omitted]
 3 | 1519 | 2746 | 960 | 318 | 38±15 | 209±66 | 0.23±0.01 | 2.1×10$^7$ | | 8.0×10$^5$ |
| | June 5 | 1810 | 5809 | 878 | 502 | 35±11 | 875±288 | 0.46±0.07 | 6.67×10$^7$ | 46 | 6.5×10$^5$ |
| | June 12 | 2554 | 9664 | 1870 | 1103 | 85±17 | 877±277 | 0.57±0.09 | 3.61×10$^7$ | | 3.4×10$^5$ |
| Idle-Biodiesel-None | June 4 | 1118 | 4270 | 867 | 344 | 22±12 | 999±316 | 0.52±0.07 | 9.17×10$^7$ | 46[@] | 7.3×10$^{5@}$ |
| | June 8 | 2160 | 8169 | 1578 | 811 | 69±20 | 1415±468 | 0.36±0.03 | 4.72×10$^7$ | | 4.1×10$^{5@}$ |
| Load-Diesel-None | June 3 | 959 | 1558 | 4999 | 379 | 19±11 | 181±58 | 0.37±0.01 | 3.6×10$^7$ | 190 | 5.3×10$^5$ |
| | June 5 | 711 | 1400 | 6690 | 34 | 37±13 | 253±100 | 0.32±0.04 | 2.61×10$^7$ | | 4.4×10$^5$ |
| Load- Biodiesel-None | June 4 | 1634 | 1410 | 6364 | 30 | 29±18 | 645±204 | 0.38±0.05 | 2.78×10$^7$ | 190[@] | 5.0×10$^{5@}$ |
| | June 8 | 518 | 2051 | 10813 | 496 | 46±22 | 284±106 | 0.30±0.04 | 1.42×10$^7$ | | 3.3×10$^{5@}$ |
| Idle-Diesel-DPF+DOC | June 9 | 2135 | 7473 | 2383 | 23 | 1.5±0.6 | 1040±335 | 0.37±0.02 | 5×10$^7$ | 52 | 910 |
| Load-Diesel-DPF+DOC | June 9 | 303 | 85 | 6157 | 4483 | 1.6±3.6 | 146±48 | 0.29±0.01 | 1.31×10$^7$ | 57 | 968 |
| Idle-Biodiesel-DPF+DOC | June 10 | 1773 | 7452 | 2213 | 182 | 2.6±1 | 787±250 | 0.44±0.04 | 5.28×10$^7$ | 52[@] | 910[@] |
| Load-Biodiesel-DPF+DOC | June 10 | 261 | 58 | 5475 | 4525 | 2±0.14 | 107±9 | 0.29±0.01 | 1.39×10$^7$ | 57[@] | 968[@] |

[revised manuscript text omitted]